# CONVERGENCE OF BAYESIAN BILEVEL OPTIMIZATION

**Shi Fu**[1]  **Fengxiang He**[2]  **Xinmei Tian**[1]  **Dacheng Tao**[3]
[1]University of Science and Technology of China,  [2]University of Edinburgh,
[3]Nanyang Technological University
`fs311@mail.ustc.edu.cn`, `F.He@ed.ac.uk`, `xinmei@ustc.edu.cn`,
`dacheng.tao@gmail.com`

## ABSTRACT

This paper presents the first theoretical guarantee for Bayesian bilevel optimization (BBO) that we term for the prevalent bilevel framework combining Bayesian optimization at the outer level to tune hyperparameters, and the inner-level stochastic gradient descent (SGD) for training the model. We prove sublinear regret bounds suggesting simultaneous convergence of the inner-level model parameters and outer-level hyperparameters to optimal configurations for generalization capability. A pivotal, technical novelty in the proofs is modeling the excess risk of the SGD-trained parameters as evaluation noise during Bayesian optimization. Our theory implies the inner unit horizon, defined as the number of SGD iterations, shapes the convergence behavior of BBO. This suggests practical guidance on configuring the inner unit horizon to enhance training efficiency and model performance.

## 1 INTRODUCTION

Hyperparameter optimization is a crucial step in the practical implementation of deep learning. Inappropriate hyperparameter configurations can lead to poor model performance and ineffective utilization in real-world systems. Bayesian optimization, relying on probabilistic models, is an effective approach for hyperparameter tuning (Snoek et al. (2012)). It can identify near-optimal configurations within a few iterations, even in non-convex and derivative-free scenarios (Frazier (2018)). However, it often focuses solely on hyperparameters while neglecting model parameters. In contrast, bilevel optimization furnishes an alternative framework that concurrently optimizes hyperparameters and model parameters in a unified architecture (Colson et al. (2007); Franceschi et al. (2018)). Specifically, bilevel optimization nests an inner-level problem of model parameter optimization within an outer-level problem of tuning hyperparameters (Bao et al. (2021).

Bayesian bilevel optimization, merging outer-level hyperparameter tuning with inner-level model parameter optimization via SGD, shows significant promise in engineering applications. Notably, the most common application is training neural network parameters with inner SGD, while tuning critical hyperparameters like learning rates and layer widths with outer Bayesian optimization (Nguyen et al. (2020);Dewancker et al. (2016);Snoek et al. (2015)). However, the underlying working mechanisms and theoretical convergence guarantees of this approach remain unclear. Additionally, properly configuring the inner unit horizon presents challenges. Limited inner SGD iterations can impede model convergence, reducing the accuracy of outer Bayesian optimization evaluations. Thus, more Bayesian iterations may be needed to ensure hyperparameter convergence. However, excessive SGD iterations waste resources, revealing a trade-off between model training and hyperparameter tuning iterations (Hasanpour et al. (2016);Li et al. (2017)). While multi-fidelity Bayesian optimization also examines this trade-off ( Nguyen et al. (2020)), it focuses more on utilizing fidelities to reduce optimization costs. Conversely, our objective is to deduce the optimal inner unit horizon that enables simultaneous convergence of model parameters and hyperparameters to generalization optima.

At the outer layer, Bayesian optimization is adopted for hyperparameter tuning. We utilize two acquisition functions: Expected Improvement (EI) and Upper Confidence Bound (UCB). Regarding EI, Bull (2011) studied its convergence in noise-free settings, while Gupta et al. (2022) recently provided convergence analysis in noisy settings with the standard predictive mean incumbent. For UCB, Srinivas et al. (2010) first introduced the method and derived regret bounds scaling as $\mathcal{O}(\sqrt{T\gamma_T})$,

where $\gamma_T$ denotes the maximum information gain between $T$ observations and the Gaussian process model. Chowdhury & Gopalan (2017) further improved it through an enhanced algorithm.

Previous theoretical studies have relied on certain noise assumptions, including bounded (Srinivas et al. (2010); Nguyen & Gupta (2017)) or sub-Gaussian noise (Chowdhury & Gopalan (2017); Gupta et al. (2022)). However, these assumptions do not adequately capture the intricacies of integrating SGD and Bayesian optimization in a bilevel framework. A key limitation in most prior works is their modeling of the noise $\{\varepsilon_t\}_{t=1}^T$ as a martingale difference sequence. This approach does not align with hyperparameter optimization, where the objective function often represents generalization performance (Snoek et al. (2012)). Another drawback of prior works (Srinivas et al. (2010); Chowdhury & Gopalan (2017)) is the use of UCB balancing coefficients that increase rapidly, which has been shown to be over-exploratory for many practical problems (De Ath et al. (2021)).

To tackle these challenges, a pivotal innovation of our work involves modeling the excess risk of inner-level SGD-trained parameters as the primary noise source during outer-level Bayesian optimization. This perspective facilitates more adaptable convergence guarantees for the bilevel setting. Modeling the SGD excess risk is critical since it builds a connection between the analytical frameworks of Bayesian optimization and bilevel optimization. The distinct objectives and assumptions of these two frameworks pose difficulties in ensuring theoretical convergence. Our approach bridges this gap, enabling a comprehensive convergence analysis for BBO.

By modeling the excess risk of SGD as noise, we derive sublinear regret bounds for BBO, guaranteeing simultaneous convergence of model parameters and hyperparameters to optimal configurations for generalization capability. This is accomplished through carefully configuring the inner unit horizon. By determining the SGD-to-Bayesian iteration trade-off, we optimize this ratio to improve computational efficiency. Furthermore, we optimize the UCB balancing coefficients based on the iteration ratio, enhancing flexibility and mitigating excessive exploration resulting from rapidly escalating coefficients in previous works. Our analyses provide valuable insights into balancing computational resources in engineering applications. The main contributions of this work include:

1. We provide a novel theoretical analysis of convergence guarantees for generalization performance within a BBO framework. A key technical innovation is modeling the excess risk of SGD-trained parameters as evaluation noise in outer-level Bayesian optimization. This tackles theoretical analysis challenges by bridging the analytical frameworks of Bayesian optimization and bilevel optimization.

2. We establish a regret bound of $\mathcal{O}(\sqrt{T\gamma_T})$ for BBO using the EI function, with a noise assumption more aligned with practical scenarios. Our regret bound achieves savings of $\sqrt{\gamma_T}$ compared to Gupta et al. (2022). This achievement is realized by optimizing the ratio of SGD iterations $N$ to Bayesian optimization iterations $T$, also offering implementation guidance.

3. We introduce adaptable balancing coefficients $\beta_t$ for the UCB acquisition function. Through this adaptation, we establish a sublinear regret bound for BBO with the UCB that holds even with fewer SGD steps, enhancing the flexibility of the inner unit horizon. Additionally, we overcome the limitations of rapidly increasing coefficients from previous analyses.

## 2    RELATED WORK

Hyperparameter optimization is crucial for leveraging deep learning's capabilities (Yang & Shami (2020); Elsken et al. (2019)). Techniques span Bayesian optimization (Wu et al. (2019); Victoria & Maragatham (2021)), decision theory (Bergstra & Bengio (2012)), multi-fidelity methods (Li et al. (2017)), and gradient-based approaches (Maclaurin et al. (2015)). We focus on BBO, exploring Bayesian optimization and bilevel frameworks' relevant aspects for hyperparameter tuning.

**Bayesian optimization**. Bayesian optimization (BO) (Osborne & Osborne (2010); Kandasamy et al. (2020)) is a prevalent approach for hyperparameter tuning by efficiently exploring and exploiting hyperparameter spaces (Nguyen et al. (2017)). Gaussian processes (Bogunovic et al. (2018)) are commonly used as priors in BO to model uncertainty and estimate objective function distributions (Bro (2010); Wilson et al. (2014)). Among acquisition functions guiding queries in BO, the EI acquisition function (Jones & Welch (1998); Malkomes & Garnett (2018); Scarlett et al. (2017); Qin et al. (2017)) is one of the most widely utilized for balancing exploration-exploitation (Nguyen & Osborne (2020); Zhan & Xing (2020)). Other acquisitions like UCB (Valko et al. (2013)), knowledge

gradient (Frazier et al. (2009); Scott et al. (2011)), Thompson sampling (Chowdhury & Gopalan (2017)), and predictive entropy search (Hernández-Lobato et al. (2014)) cater to various scenarios.

Theoretical analyses of Bayesian optimization have been conducted to understand their convergence properties (Ryzhov (2016);Gupta et al. (2020)). The EI has been studied under specific conditions (Vazquez & Bect (2010); Bull (2011); Gupta et al. (2021)). Adapting EI for noisy evaluations poses challenges. Wang & de Freitas (2014) addressed this issue by introducing an alternative incumbent selection criterion that necessitates an additional global optimization step. Nguyen & Gupta (2017) used the best-observed value as the incumbent, showing sublinear convergence with a minimum improvement stopping condition. Gupta et al. (2022) analyzed EI convergence with the standard predictive mean incumbent. For UCB, Srinivas et al. (2010) established regret bound as $\mathcal{O}(\sqrt{T\gamma_T})$. This approach was subsequently enhanced by ( Janz et al. (2020)) via a modified UCB algorithm.

**Bilevel optimization**. Bilevel optimization offers an alternative framework for hyperparameter tuning by concurrently optimizing hyperparameters and model parameters in a nested architecture (Colson et al. (2007);Franceschi et al. (2018);Bao et al. (2021)). These methods encompass implicit gradient (Bengio (2000);Luketina et al. (2016);Pedregosa (2016);Lorraine et al. (2020)), hypernetworks (Lorraine & Duvenaud (2018);MacKay et al. (2019)), unrolled differentiation (Franceschi et al. (2017);Maclaurin et al. (2015);Bao et al. (2021)), and cross-validation (Bergstra & Bengio (2012)). Implicit gradient-based methods estimate the model's performance gradient with respect to hyperparameters without explicit computation (Lorraine et al. (2020)). Hypernetworks generate model weights conditioned on hyperparameters. Cross-validation approximates optimal hyperparameters using grid search (Abas et al. (2020)) or random search (Bergstra & Bengio (2012)). Unrolled differentiation involves optimizing the inner problem over multiple iterations (Fu et al. (2016)).

**Remark**. While Bayesian and bilevel optimization have advanced theoretically, extending convergence guarantees to BBO remains challenging. Bayesian optimization analyses often disregard the impact of model parameter convergence on noise during evaluations, instead relying on unrealistic noise assumptions. Theoretical bilevel optimization analyses struggle to extend to the non-convex, nonlinear, and derivative-free scenarios inherent to Bayesian optimization.

## 3 PRELIMINARIES

We consider bilevel optimization involving model parameters and hyperparameters. Let $\mathcal{Z}$, $\Theta$, and $\Lambda$ denote the data space, the parameter space, and the hyperparameter space, respectively. Given hyperparameters $\lambda \in \Lambda$, model parameters $\theta \in \Theta$ and distribution $\mathcal{D}$ on the data space $\mathcal{Z}$, the expected error $L(\lambda, \theta)$ is defined as $\mathbb{E}_{z \sim \mathcal{D}}[\ell(\lambda, \theta, z)]$, where $\ell : \Lambda \times \Theta \times \mathcal{Z} \to \mathbb{R}$ is the bounded loss function. Our objective is to find the hyperparameter-model parameter pair that minimizes the expected error over an unknown distribution. This objective entails nesting two search problems:

$$\lambda^* = \arg\min_{\lambda \in \Lambda} L(\lambda, \theta_\lambda^*), \text{ where } \theta_\lambda^* = \arg\min_{\theta \in \Theta} L(\lambda, \theta). \tag{3.1}$$

The inner-level objective is to determine the optimal model parameters, denoted as $\theta_\lambda^*$, for a given hyperparameter $\lambda$. It's worth noting that the value of $\theta_\lambda^*$ depends on the choice of the hyperparameter $\lambda$. At the outer level, the goal is to identify the optimal hyperparameter $\lambda^*$, which determines its associated model parameters $\theta_{\lambda^*}^*$, collectively minimizing the expected error. This bilevel structure is very classic and has been applied in many practical scenarios, such as in Darts (Liu et al. (2018)).

**Regret**. Similar to Gupta et al. (2022), we employ regret as a convergence measure. We define $\lambda^*$ and $\theta_{\lambda^*}^*$ as the global minimum points, as shown in Equation 3.1. During the $t$-th iteration at the outer level, the output hyperparameters are denoted as $\lambda_t^+$. Simultaneously, at the inner level, using these hyperparameters $\lambda_t^+$, we optimize the model parameters through $N$ steps of SGD, denoted as $\theta_{\lambda_t^+}^N$. The cumulative regret $R_T$ at iteration $T$ is calculated as the sum of instantaneous regrets: $R_T = \sum_{t=1}^{T} r_t$, where $r_t = L(\lambda^*, \theta_{\lambda^*}^*) - L(\lambda_t^+, \theta_{\lambda_t^+}^N)$. Our objective is to demonstrate that $R_T$ exhibits almost sublinear growth, specifically $\lim_{T \to \infty} R_T/T = 0$.

### 3.1 INNER LEVEL OF BBO: MODEL TRAINING BY SGD

At the inner level, our goal is to optimize the model parameters $\theta$ by minimizing the expected error. Given the limited knowledge of data distribution, direct access to this expected error is challenging.

As a solution, we utilize SGD to optimize the empirical error, serving as a proxy for improving model parameters. Previous studies, such as Hardt et al. (2016), have shown SGD's effectiveness in achieving strong generalization performance. Let $S^{tr} = \{z_i^{tr}\}_{i=1}^n$ denote the training dataset, with $n$ representing the number of training samples. Formally, the inner-level problem is defined as: $\min_{\theta \in \Theta} L^{tr}(\lambda, \theta, S^{tr}) = \frac{1}{n} \sum_{i=1}^n \ell(\lambda, \theta, z_i^{tr})$. Below, we present some essential definitions.

**Definition 1.** (Lipschitz, smoothness, and convexity). Let constants $K, \gamma > 0$. Consider the function $\ell : \Lambda \times \Theta \times \mathcal{Z} \to \mathbb{R}$. We define the following properties:

- **Lipschitz Continuity**: The loss $\ell(\lambda, \theta, z)$ is said to be $K$-Lipschitz continuous with respect to $\theta$ if $\|\ell(\lambda, \theta_1, z) - \ell(\lambda, \theta_2, z)\| \leq K\|\theta_1 - \theta_2\|$ for any $\theta_1, \theta_2, z, \lambda$.

- **Smoothness**: The loss $\ell(\lambda, \theta, z)$ is said to be $\gamma$-Smooth with respect to $\theta$ if $\|\nabla_\theta \ell(\lambda, \theta_1, z) - \nabla_\theta \ell(\lambda, \theta_2, z)\| \leq \gamma\|\theta_1 - \theta_2\|$ for any $\theta_1, \theta_2, z, \lambda$.

- **Convexity**: The loss $\ell(\lambda, \theta, z)$ is said to be convex with respect to $\theta$ if $\ell(\lambda, \theta_1, z) \geq \ell(\lambda, \theta_2, z) + \langle \nabla_\theta \ell(\lambda, \theta_2, z), \theta_1 - \theta_2 \rangle$ for any $\theta_1, \theta_2, z, \lambda$.

The above definitions are standard concepts that have been widely adopted in related works such as Ghadimi & Wang (2018), Ji & Liang (2023), Grazzi et al. (2020), and Ji et al. (2021).

**Remark 1.** Our BBO theoretical framework can be extended to non-smooth and non-convex situations. Emphasizing BBO convergence, we've mentioned these concepts for clarity. The proposed structure is highly adaptable for various scenarios. Drawing on insights from Charles & Papailiopoulos (2018), the framework extends to non-convex situations under a $\mu$-gradient-dominance condition. For non-convex conditions, Theorem 1 indicates an excess risk of $\mathcal{O}\left((N^{-1/2}\mu^{-1/2} + N^{-1/4}\mu^{-3/4})\log N\right)$. In addition, we eliminate the smoothness assumption, adding a term $\eta\sqrt{N}$ to the excess risk bound, where $\eta$ is the step size. Alternatively, relaxing the assumption to $\alpha$-Hölder continuity introduces an additional term $\mathcal{O}\left(N^{1-\frac{1}{2(1-\alpha)}}\log N\right)$ to the excess risk bound. The detailed content is in Appendix F.

## 3.2 Outer Level of BBO: Hyperparameter Tuning by BO

At the outer level, the model parameters $\theta_\lambda^*$ remain fixed, and our objective is to identify the hyperparameters $\lambda^* \in \Lambda$ that minimize the expected error $L(\lambda, \theta_\lambda^*)$. Critically, the function $L(\lambda, \theta_\lambda^*)$ has a uniquely determined value for each $\lambda$. This uniqueness arises because, although $\theta_\lambda^* = \arg\min_{\theta \in \Theta} L(\lambda, \theta)$ might represent a set when $\lambda$ is specified, for any $\theta_\lambda^*$ in this set, the loss function $L(\lambda, \theta_\lambda^*)$ consistently yields the unique value $\min_{\theta \in \Theta} L(\lambda, \theta)$. Therefore, we may treat $L(\lambda, \theta_\lambda^*)$ as a function solely dependent on $\lambda$, allowing us to conceptualize it as a mapping function from $\Lambda$ to $\mathbb{R}$.

### 3.2.1 Regularity Assumptions

As noted by Scarlett et al. (2017), Bayesian optimization can be viewed from two distinct perspectives: Bayesian and non-Bayesian. Departing from the standard Bayesian optimization framework, this work adopts a more agnostic, non-Bayesian setting (Srinivas et al. (2010)). Specifically, we assume that hyperparameter space $\Lambda$ is a compact set. We also posit the reward function $L(\lambda, \theta_\lambda^*)$ lies within a reproducing kernel Hilbert space (RKHS) $\mathcal{H}_k(\Lambda)$ of functions mapping from $\Lambda$ to $\mathbb{R}$, characterized by a positive semi-definite kernel function $k : \Lambda \times \Lambda \to \mathbb{R}$. This RKHS $\mathcal{H}_k(\Lambda)$ is determined by its kernel function $k(\cdot, \cdot)$ and features an inner product $\langle \cdot, \cdot \rangle_k$ satisfying the reproducing property: $L(\lambda, \theta_\lambda^*) = \langle L(\cdot, \theta_\cdot^*), k(\lambda, \cdot) \rangle_k$ for all $L(\cdot, \theta_\cdot^*) \in \mathcal{H}_k(\Lambda)$. Hence, the kernel $k$ effectively represents the evaluation map at each $\lambda \in \Lambda$ through the RKHS inner product. The RKHS norm, defined as $\|L(\cdot, \theta_\cdot^*)\|_k = \sqrt{\langle L(\cdot, \theta_\cdot^*), L(\cdot, \theta_\cdot^*) \rangle_k}$. We assume the RKHS norm of the unknown $L(\cdot, \theta_\cdot^*)$ is bounded by $B$. This assumption suggests the smoothness of the function. This relationship is showcased in the following example that the inequality $|L(\lambda_1, \theta_{\lambda_1}^*) - L(\lambda_2, \theta_{\lambda_2}^*)| = |\langle L(\cdot, \theta_\cdot^*), k(\lambda_1, \cdot) - k(\lambda_2, \cdot) \rangle| \leq \|L(\cdot, \theta_\cdot^*)\|_k \|k(\lambda_1, \cdot) - k(\lambda_2, \cdot)\|_k$ by the Cauchy-Schwarz inequality.

### 3.2.2 Gaussian Process (GP) Regression

Bayesian optimization employs GPs as flexible surrogate models for the objective function, leveraging their versatile priors and tractable posteriors (Rasmussen & Williams (2006)). Initially, we assume

a GP prior, denoted as $GP_\Lambda(0, k(\cdot, \cdot))$, for the unknown reward function $L(\cdot, \theta^*)$ over $\Lambda$, with $k(\cdot, \cdot)$ representing the kernel function associated with the RKHS $\mathcal{H}_k(\Lambda)$. The algorithm collects observations $\boldsymbol{y}_t = [y_1, \ldots, y_t]^T$ at points $A_t = \{\lambda_1, \ldots, \lambda_t\}$, where $y_t = L(\lambda_t, \theta^*_{\lambda_t}) + \varepsilon_t$, and $\varepsilon_t \sim N(0, \sigma^2)$ denotes independent and identically distributed Gaussian noise for all $t$. Consequently, given the history $H_{1:t} = \{\lambda_i, y_i\}_{i=1}^t$, let $\sigma_t^2(\lambda) = k_t(\lambda, \lambda)$, the posterior distribution over $L(\cdot, \theta^*)$ remains a GP, denoted as $P(L_t(\cdot, \theta^*) \mid H_{1:t-1}, \lambda) = \mathcal{N}(\mu_t(\lambda), \sigma_t^2(\lambda))$, where

$$\mu_t(\lambda) = \boldsymbol{k}_t(\lambda)^T (\boldsymbol{K}_t + \sigma^2 \boldsymbol{I})^{-1} \boldsymbol{y}_t, \quad k_t(\lambda, \lambda') = k(\lambda, \lambda') - \boldsymbol{k}_t(\lambda)^T (\boldsymbol{K}_t + \sigma^2 \boldsymbol{I})^{-1} \boldsymbol{k}_t(\lambda'),$$

where we define $\boldsymbol{k}_t(\lambda) = [k(\lambda_1, \lambda) \ldots k(\lambda_t, \lambda)]^T$, $\boldsymbol{K}_t = [k(\lambda, \lambda')]_{\lambda, \lambda' \in A_t}$. We assume the kernel function $k$ is fixed and well-defined with $k(x, x) \leq 1$.

### 3.2.3 ACQUISITION FUNCTION

Acquisition functions are vital in Bayesian optimization as they guide the choice of the next evaluation point. They quantify the potential improvement of each candidate point. This section explores two prevalent acquisition functions: UCB (Srinivas et al. (2010)) and EI (Jones & Welch (1998)).

**Upper Confidence Bound**. The UCB acquisition balances exploration and exploitation by selecting points with potential improvement while considering evaluation uncertainty. It is defined as:

$$UCB_t(\lambda) = -\mu_{t-1}(\lambda) + \beta_t \sigma_{t-1}(\lambda), \tag{3.2}$$

where $\beta_t$ denotes coefficients regulating the trade-off. The $-\mu_{t-1}(\lambda)$ term promotes exploitation by favoring points with lower estimated mean values, thereby increasing the likelihood of improved outcomes. The $\sigma_{t-1}(\lambda)$ term drives exploration by prioritizing highly uncertain points.

**Expected Improvement**. The EI quantifies the expected enhancement in the incumbent relative to the objective function value. Typically, the incumbent is chosen as the minimum GP mean observed so far, denoted as $\mu_t^+ = \min_{\lambda_i \in A_t} \mu_{t-1}(\lambda_i)$, consistent with Gupta et al. (2022). The EI is defined as $EI_t(\lambda) = \mathbb{E}\left[\max\{\mu_t^+ - L(\lambda, \theta^*_\lambda), 0\} \mid H_{1:t}\right]$. The closed-form expression for EI can be derived as:

$$EI_t(\lambda) = (\mu_t^+ - \mu_{t-1}(\lambda))\Phi\left(\frac{\mu_t^+ - \mu_{t-1}(\lambda)}{\sigma_{t-1}(\lambda)}\right) + \sigma_{t-1}(\lambda)\phi\left(\frac{\mu_t^+ - \mu_{t-1}(\lambda)}{\sigma_{t-1}(\lambda)}\right), \text{ if } \sigma_{t-1}(\lambda) > 0,$$

where $\phi$ is the standard normal p.d.f. and $\Phi$ is the c.d.f. When $\sigma_{t-1}(\lambda) = 0$, we set $EI_t(\lambda) = 0$.

## 4 MAIN RESULTS

This section presents theoretical analyses of the convergence guarantees for BBO. Specifically, We derive an excess risk bound for the inner-level SGD-trained model parameters. We also establish regret bounds for the outer-level Bayesian optimization that ensures simultaneous convergence of model parameters and hyperparameters. The BBO algorithms are detailed in the Appendix. Based on the convergence analysis, we characterize the trade-off between model training and tuning.

### 4.1 INNER LEVEL OF BBO: EXCESS RISK BOUND

First, we present the excess risk bound for inner-level optimization.

**Theorem 1.** *Suppose that the function $\ell(\lambda, \theta, z)$ is $K$-Lipschitz continuous, $\gamma$-smooth and convex with respect to $\theta$, uniformly bounded by $M$. We perform SGD with step sizes $\eta_j = \eta \asymp \frac{1}{\sqrt{N}} \leq 2/\gamma$ on a sample $S^{tr}$ drawn from the distribution $\mathcal{D}$ at the inner level. This involves performing $N$ steps and obtaining the output result $\theta_\lambda^N$. Let $S^{val} = \{z_i^{val}\}_{i=1}^m$ represent the validation set drawn from the distribution $\mathcal{D}$. The validation error is defined as $L^{val}(\lambda, \theta_\lambda^N, S^{val}) = \frac{1}{m}\sum_{i=1}^m \ell(\lambda, \theta_\lambda^N, z_i^{val})$. Choose $N \asymp n \asymp m$. Then, with a probability of at least $1 - \delta$, we have:*

$$L^{val}(\lambda, \theta_\lambda^N, S^{val}) - L(\lambda, \theta_\lambda^*) = \mathcal{O}\left(N^{-\frac{1}{2}}\log^{3/2} N\right)$$

The complete expression is given by $L^{val}(\lambda, \theta_\lambda^N, S^{val}) - L(\lambda, \theta_\lambda^*) \leq \varphi(N)N^{-\frac{1}{2}}$, where $\varphi(N) = \mathcal{O}\left(K^2 \log N \log(1/\delta) + M\sqrt{\log(1/\delta)} + M\sqrt{\log(2/\delta)} + \log^{3/2}(N/\delta)\right)$.

**Remark 2.** In outer-level Bayesian optimization, the objective is to minimize the expected error $L(\lambda, \theta_\lambda^*)$. However, as the data distribution is unknown, $L(\lambda, \theta_\lambda^*)$ cannot be evaluated directly. Instead, the validation loss $L^{val}(\lambda, \theta_\lambda^N, S^{val})$ serves as a proxy to observe $L(\lambda, \theta_\lambda^*)$. Specifically, given hyperparameters $\lambda$ and model parameters $\theta_\lambda^N$ from inner-level SGD, the difference between $L^{val}(\lambda, \theta_\lambda^N, S^{val})$ and $L(\lambda, \theta_\lambda^*)$ represents noise when evaluating $L(\lambda, \theta_\lambda^*)$ in outer optimization. Analyzing the convergence of this excess risk bound $L^{val}(\lambda, \theta_\lambda^N, S^{val}) - L(\lambda, \theta_\lambda^*)$ is of interest.

## 4.2 OUTER LEVEL OF BBO: REGRET BOUND

This section presents regret bounds for the outer-level Bayesian optimization. The subsequent lemma delineates the concentration of the posterior mean around the true objective function, thereby constituting the theoretical basis for the ensuing regret analysis. Notably, as explicated in Section 3.2, $L(\lambda, \theta_\lambda^*)$ may be considered as functions mapping from $\Lambda$ to $\mathbb{R}$.

**Lemma 2.** *let* $L(\cdot, \theta_\cdot^*) : \Lambda \to \mathbb{R}$ *be a member of the RKHS of real-valued functions on $\Lambda$ with kernel $k$, with RKHS norm bounded by $B$. Consider the noise term $\varepsilon_t = L^{val}\left(\lambda_t, \theta_{\lambda_t}^N, S^{val}\right) - L\left(\lambda_t, \theta_{\lambda_t}^*\right)$, then, with probability at least $1 - \delta$, the following holds for all $\lambda \in \Lambda$ and $t \geq 1$:*

$$|\mu_t(\lambda) - L(\lambda, \theta_\lambda^*)| \leq \sqrt{B^2 + \sigma^{-2}t\varphi^2(N)N^{-1}}\sigma_t(\lambda).$$

Subsequently, we introduce the definition of maximum information gain, which will prove useful in bounding the cumulative regret.

**Definition 2.** Given the set $A_T = \{\lambda_1, \cdots, \lambda_T\} \subset \Lambda$, let $\boldsymbol{L}_{A_T}(\cdot, \theta_\cdot^*) = [L(\lambda, \theta_\lambda^*)]_{\lambda \in A_T}$ and $\boldsymbol{y}_{A_T} = \boldsymbol{L}_{A_T}(\cdot, \theta_\cdot^*) + \boldsymbol{\varepsilon}_{A_T}$, where $\boldsymbol{\varepsilon}_{A_T} \sim N\left(\boldsymbol{0}, \sigma^2\boldsymbol{I}\right)$. With $I(\cdot\,;\cdot)$ denoting the mutual information, the maximum information gain after $T$ iterations is defined as $\gamma_T = \max_{A_T \in \Lambda} I\left(\boldsymbol{y}_{A_T}; \boldsymbol{L}_{A_T}(\cdot, \theta_\cdot^*)\right)$.

Notably, the maximum information gain $\gamma_T$ differs under various kernel selections. According to the results from Srinivas et al. (2010), for the Squared Exponential kernel, $\gamma_T = \mathcal{O}(\log^{d+1}(T))$, whereas based on the findings in Vakili et al. (2021), for the Matérn-$\nu$ kernel, $\gamma_T = \widetilde{\mathcal{O}}(T^{\frac{d}{2\nu+d}})$.

### 4.2.1 REGRET BOUND FOR BBO WITH EI FUNCTION

In the ensuing analysis, we establish the primary theoretical contribution of this work, specifically, the regret bound for Bayesian bilevel optimization with the EI acquisition function. For the sake of clarity in notation, we define the auxiliary function $\tau(z) = z\Phi(z) + \phi(z)$.

**Theorem 3.** *Let $\delta \in (0, 1)$. Assume that the true underlying function $L(\lambda, \theta_\lambda^*)$ in terms of $\lambda$ lies in the RKHS $\mathcal{H}_k(\Lambda)$ associated with the kernel $k(\lambda, \lambda')$. Furthermore, consider the noise term $\varepsilon_t = L^{val}\left(\lambda_t, \theta_{\lambda_t}^N, S^{val}\right) - L\left(\lambda_t, \theta_{\lambda_t}^*\right)$. Specifically, assume that $\|L(\cdot, \theta_\cdot^*)\|_k \leq B$ and define $\beta_t = \sqrt{B^2 + \sigma^{-2}t\varphi^2(N)N^{-1}}$. By executing algorithm 1 with EI acquisition and the prior $GP_\Lambda\left(0, k(\cdot, \cdot)\right)$, with probability at least $1 - \delta$, the cumulative regret is bounded as:*

$$R_T = \mathcal{O}\left(\frac{\beta_T^2\sqrt{T\gamma_T}}{\tau(\beta_T) - \beta_T} + TN^{-\frac{1}{2}}\right),$$

*If we select $N \asymp T$, we attain:*

$$R_T = \mathcal{O}\left(\sqrt{T\gamma_T}\right)$$

**Proof Sketch of Theorem 3**. Adapting Bayesian regret analysis to the bilevel optimization context with noise induced by SGD poses a central challenge. To bound the cumulative regret $R_T$, the initial step involves constraining the instantaneous regrets $r_t$, defined as $r_t = L\left(\lambda_t^+, \theta_{\lambda_t^+}^N\right) - L(\lambda^*, \theta_{\lambda^*}^*)$. Each instantaneous regret $r_t$ can be decomposed into three terms: $r_t = L\left(\lambda_t^+, \theta_{\lambda_t^+}^N\right) - L\left(\lambda_t^+, \theta_{\lambda_t^+}^*\right) + L\left(\lambda_t^+, \theta_{\lambda_t^+}^*\right) - \mu_t^+ + \mu_t^+ - L(\lambda^*, \theta_{\lambda^*}^*)$, where $\mu_t^+$ denotes the incumbent.

The first term, $L\left(\lambda_t^+, \theta_{\lambda_t^+}^N\right) - L\left(\lambda_t^+, \theta_{\lambda_t^+}^*\right)$, can be bounded by applying concentration inequalities and the excess risk result from Theorem 1, yielding an upper bound of $\mathcal{O}(N^{-1/2} \log N)$. For the

second term, $L\left(\lambda_t^+, \theta_{\lambda_t^+}^*\right) - \mu_t^+$, the bound is given by $\beta_t \left(\sqrt{2\pi} \left(\beta_t + 1\right) \sigma_{t-1}\left(\lambda_t\right) + \sqrt{2\pi} I_t\left(\lambda_t\right)\right)$, where $I_t(\lambda_t) = \max \left\{\mu_t^+ - L\left(\lambda_t, \theta_{\lambda_t}^*\right), 0\right\}$. To bound the last term $\mu_t^+ - L\left(\lambda^*, \theta_{\lambda^*}^*\right)$, our approach involves initially establishing lower and upper bounds on the EI acquisition value by leveraging the properties of $\tau(z)$. A pivotal technical challenge arises in bounding the discrepancy between the GP posterior mean $\mu_t(\cdot)$ and the true function value $L(\cdot, \theta^*)$ at $\lambda_t$ and $\lambda^*$, which requires considering the SGD-induced noise as detailed in Lemma 2. The optimality of $\lambda_t$ from maximizing the EI acquisition leads to an upper bound given by: $(\frac{1+\beta_t}{\tau(\beta_t)-\beta_t})(I_t(\lambda_t) + (\beta_t + 1)\sigma_{t-1}(\lambda_t))$, as shown in Lemma 15.

Bounding and summing the three constituent terms allows us to derive the cumulative regret $R_T$. Employing auxiliary results related to the cumulative sums of improvement terms $I_t(\lambda_t)$ and predictive variances $\sigma_t(\lambda_t)$, as documented in Gupta et al. (2022) and Chowdhury & Gopalan (2017), and setting $N \asymp T$, we attain the bound $R_T = \mathcal{O}(\sqrt{T\gamma_T})$.

**Comparision with previous works**. Through appropriate selection of $N$, our regret bound is denoted as $R_T = \mathcal{O}(\sqrt{T\gamma_T})$. In contrast, Nguyen & Gupta (2017) established a regret bound of $\mathcal{O}(\gamma_T \sqrt{T \log T})$ for EI with the best-observed value $y^+$ as the incumbent. They further introduced an additional assumption constraining the variance function values to exceed a lower bound $\kappa$. Thus, their bound escalates markedly as $\kappa \to \infty$. The work most analogous to ours is Gupta et al. (2022), which also utilizes EI with the standard incumbent $\mu_t^+ = \min_{\lambda_i \in A_t} \mu_{t-1}(\lambda_i)$, and possesses regret bounds of $R_T = \mathcal{O}(\gamma_T \sqrt{T})$. Our regret bound thus achieves theoretical savings of $\sqrt{\gamma_T}$.

Furthermore, It's worth noting that Gupta et al. (2022) assumed conditional $R$-sub-Gaussian noise, while Nguyen & Gupta (2017) assumed bounded noise. Additionally, both studies assumed that the noise sequence $\{\varepsilon_t\}_{t=1}^T$ constitutes a martingale difference sequence, signifying that $\mathbb{E}[\varepsilon_t|\varepsilon_{<t}] = 0$ for all $t \in T$. In comparison, our assumption regarding the noise $\varepsilon_t$, denoted as the excess risk of the inner-level SGD-optimized model parameters, is $\varepsilon_t = L^{val}(\lambda_t, \theta_{\lambda_t}^N, S^{val}) - L(\lambda_t, \theta_{\lambda_t}^*)$. The conditional expectation $\mathbb{E}\left[\varepsilon_t \mid \varepsilon_{<t}\right] = 0$ does not hold here. This is because $\theta_{\lambda_t}^N$ depends on the training set $S^{tr}$ while $\theta_{\lambda_t}^*$ does not. This modeling more aligns closely with practical scenarios when leveraging Bayesian optimization for hyperparameter tuning.

Moreover, prior research has primarily restricted its regret definition to hyperparameters. In contrast, as outlined in Section 3, our approach uniquely defines regret to include both model parameters and hyperparameters. This broader view adds complexity to our theoretical analysis by necessitating the establishment of a sublinear regret bound. Our derived bound offers theoretical validation for the convergence of both model parameters and hyperparameters towards their optimal values.

**Practical insights: Inner Unit Horizon as the Same Order of Outer-Level Iterations.**. By judiciously selecting the number of inner-level SGD optimization iterations as $N \asymp T$, we attained the regret bound $R_T = \mathcal{O}(\sqrt{T\gamma_T})$. This further provides practical guidance into better balancing the iterations between inner-level SGD and outer-level Bayesian optimization.

Specifically, if the number of SGD iterations N is substantially less than $T$, such as $\mathcal{O}(\sqrt{T})$, the regret becomes $R_T = \mathcal{O}(T\sqrt{\gamma_T}/(\tau(T^{1/4}) - T^{1/4}))$. Since $\gamma_T$ monotonically increases and $\tau(z) - z$ monotonically decreases to zero, when $T \to \infty$, we have $\gamma_T \to \infty$ and $\tau(T^{1/4}) - T^{1/4} \to 0$. Consequently, $R_T/T = \mathcal{O}(\sqrt{\gamma_T}/(\tau(T^{1/4}) - T^{1/4}))$ will exhibit rapid growth. Indeed, this suggests that having too few inner SGD iterations results in a faster increase in the regret bound. In contrast, if $N$ significantly exceeds $T$, then $\beta_T \to B^2$, maintaining a constant order instead of diminishing. Hence, the regret $R_T = \mathcal{O}(\sqrt{T\gamma_T})$ remains unchanged, indicating unnecessary SGD iterations being performed and wasted resources.

In summary, the optimal inner unit horizon $N$ for convex functions scales as $N \asymp T$. Moreover, we demonstrate in Section F.1 that for non-convex functions, the optimal $N \asymp T^2$. This choice of $N$ strikes an effective balance between SGD and Bayesian optimization iterations.

### 4.2.2 REGRET BOUND FOR BBO WITH THE UCB FUNCTION

Next, we establish the regret bound for the Bayesian bilevel optimization with the UCB acquisition.

**Theorem 4.** *Let $\delta \in (0, 1)$. Assume that the true underlying function $L(\lambda, \theta_\lambda^*)$ in terms of $\lambda$ lies in the RKHS $\mathcal{H}_k(\Lambda)$ associated with the kernel $k(\lambda, \lambda')$. Additionally, consider the noise term $\varepsilon_t = L^{val}\left(\lambda_t, \theta_{\lambda_t}^N, S^{val}\right) - L\left(\lambda_t, \theta_{\lambda_t}^*\right)$. Specifically, assume that $\|L(\cdot, \theta^*)\|_k \leq B$ and let*

$\beta_t = \sqrt{B^2 + \sigma^{-2}t\varphi^2(N)N^{-1}}$. *By running algorithm 2 with $\beta_t$ and the prior $GP_\Lambda\left(0, k(\cdot, \cdot)\right)$, with probability at least $1 - \delta$, the cumulative regret is bounded as:*

$$R_T = \mathcal{O}\left(\sqrt{(B^2 + TN^{-1})\,T\gamma_T} + TN^{-\frac{1}{2}}\right)$$

*If we select $N \asymp T$, we can obtain that:*

$$R_T = \mathcal{O}\left(\sqrt{T\gamma_T}\right)$$

**Proof Sketch of Theorem 4.** This proof approach is similar to Theorem 3, but employs a different acquisition function. The instantaneous regret $r_t$ can be decomposed into two terms: $r_t = L\left(\lambda_t^+, \theta_{\lambda_t^+}^N\right) - L\left(\lambda_t^+, \theta_{\lambda_t^+}^*\right) + L\left(\lambda_t^+, \theta_{\lambda_t^+}^*\right) - L\left(\lambda^*, \theta_{\lambda^*}^*\right)$. The first term, identical to the first term in Theorem 3, can be established using the same methodology, yielding the bound of $\mathcal{O}(N^{-1/2}\log N)$. Through the utilization of Lemma 2 and the selection of $\lambda_t^+$ as specified in Algorithm 2, we can demonstrate that the second term can be upper-bounded as follows: $2\beta_t\sigma_{t-1}\left(\lambda_t^+\right)$. By summing the upper bounds of the two terms mentioned above and taking into account that $\sum_{t=1}^T \sigma_{t-1}\left(\lambda_t^+\right) = \mathcal{O}\left(\sqrt{T\gamma_T}\right)$ as demonstrated in Chowdhury & Gopalan (2017), the final result can be deduced: $R_T = \mathcal{O}\left(\sqrt{(B^2 + TN^{-1})\,T\gamma_T} + TN^{-1/2}\right)$ with probability at least $1 - \delta$. By selecting $N \asymp T$, we can further establish that: $R_T = \mathcal{O}\left(\sqrt{T\gamma_T}\right)$.

**Comparision with previous works.** For the UCB, $\beta_t$ plays a crucial role in balancing exploration and exploitation. In previous work, Srinivas et al. (2010) introduced the GP-UCB algorithm with $\beta_t = \left(2B^2 + 300\gamma_t\log^3(t/\delta)\right)^{1/2}$, while a slightly improved variant, the IGP-UCB algorithm, presented in Chowdhury & Gopalan (2017), used $\beta_t = B + R\sqrt{2\left(\gamma_{t-1} + 1 + \ln(1/\delta)\right)}$. In both cases, the balancing coefficient $\beta_t$ tends to increase rapidly with $t$, especially when the Matérn-$\nu$ kernel is used. In practice, this leads to numerous unnecessary explorations in Bayesian optimization (De Ath et al. (2021)). However, in our work, we have chosen the balancing coefficients as $\beta_t = \sqrt{B^2 + \sigma^{-2}t\varphi^2(N)N^{-1}}$. As a result, we can appropriately adjust the ratio of the number of iterations $N$ in the inner level to the number of iterations $T$ in the outer level to control $\beta_t$.

Furthermore, by choosing $N \asymp T$ we can obtain a regret bound $\mathcal{O}\left(B\sqrt{T\gamma_T}\right)$, which is tighter than the regret bound $\mathcal{O}\left(B\sqrt{T\gamma_T} + \gamma_T\sqrt{T}\right)$ associated with Chowdhury & Gopalan (2017). Our assumptions regarding noise are also more aligned with reality. This approach enables us to achieve an improved balance between exploration and exploitation in Bayesian optimization. Moreover, our regret bound provides theoretical assurance for the convergence of both model parameters and hyperparameters to their respective optima, whereas Chowdhury & Gopalan (2017) only guarantees the convergence of hyperparameters.

**Practical insights: Flexibility of the Inner Unit Horizon.** A choice of $N \asymp T$ is also reasonable for the UCB function. If $N$ is significantly larger than $T$, the coefficient $B^2 + TN^{-1}$ tends to approach $B^2$, remaining constant rather than approaching zero, and the regret bound remains $R_T = \mathcal{O}(\sqrt{T\gamma_T})$, showing no change. This implies that inner-level SGD involves many futile iterations.

However, it's important to note that, unlike the situation for the EI acquisition function mentioned above, if $N$ is significantly smaller than $T$, for instance, $N = \mathcal{O}(\sqrt{T})$, then the regret bound becomes $R_T = \mathcal{O}(T^{3/4}\sqrt{\gamma_T})$. For the SE kernel, where $\gamma_T = \mathcal{O}(\log^{d+1}(T))$, the regret bound $R_T$ still exhibits sublinear growth. In the case of the Matérn-$\nu$ kernel, with $\gamma_T = \widetilde{\mathcal{O}}(T^{\frac{d}{2\nu+d}})$, when $2\nu > d$, the regret bound $R_T$ can also exhibit sublinear growth.

In summary, the UCB acquisition offers greater flexibility in choosing the number of iterations for inner-level SGD iterations compared to EI. With fewer SGD iterations, the noise $\varepsilon_t$ in function evaluations increases. Therefore, our theoretical analysis shows that UCB is more resilient to the influence of noise compared to EI, which aligns with the experimental findings in Hu et al. (2021).

## 5 EXPERIMENTS

We conducted numerical results in this section. In the inner level, we employ SGD to train a CNN with two convolutional layers and one fully connected layer on the MNIST dataset. In the outer level,

Bayesian Optimization uses the EI and UCB functions to adjust hyperparameters like the learning rate. We fix the number of iterations for the outer-level BO and compare the number of iterations for the inner-level SGD under different scenarios, along with their respective convergence outcomes, as detailed in the table below. We initially present the results when utilizing the EI acquisition function.

(1) Setting the number of outer Bayesian optimization steps as 20.

| SGD Iteration | 100 | 500 | 1,000 | 2,000 | 3,000 |
|---|---|---|---|---|---|
| Performance | $3.15 \pm 0.75$ | $2.73 \pm 0.03$ | $2.70 \pm 0.04$ | $2.43 \pm 0.12$ | $2.46 \pm 0.13$ |

(2) Setting the number of outer Bayesian optimization steps as 40.

| SGD Iteration | 500 | 1,000 | 2,000 | 3,000 | 4,000 |
|---|---|---|---|---|---|
| Performance | $3.45 \pm 0.85$ | $2.42 \pm 0.23$ | $2.44 \pm 0.55$ | $2.39 \pm 0.09$ | $2.51 \pm 0.09$ |

(3) Setting the number of outer Bayesian optimization steps as 60.

| SGD Iteration | 1,000 | 2,000 | 3,000 | 4,000 | 6,000 |
|---|---|---|---|---|---|
| Performance | $2.58 \pm 0.58$ | $2.40 \pm 0.35$ | $2.34 \pm 0.28$ | $2.10 \pm 0.08$ | $2.25 \pm 0.17$ |

The experiments are aligned with our theoretical analysis. Fixing the Bayesian optimization's iteration number, the loss function decreases as the SGD steps rise initially, while suboptimal hyperparameters cause high loss. Specifically, with only 20 outer-level iterations, excessively low tuning led to high loss even at high SGD steps. For 40 outer-level iterations and over 1,000 SGD steps, we see signs of overtraining and inadequate tuning. For 60-step outer-level iterations, the loss at over 4,000 SGD steps is less than in the 40-step setting, due to more adequate tuning. Yet, insufficient tuning still occurs at 6,000 SGD steps under the 60-step setting.

Then we present the results when utilizing the UCB acquisition function.

(1) Setting the number of outer Bayesian optimization steps as 20.

| SGD Iteration | 100 | 500 | 1,000 | 2,000 | 3,000 |
|---|---|---|---|---|---|
| Performance | $2.93 \pm 0.60$ | $2.59 \pm 0.47$ | $2.49 \pm 0.34$ | $2.39 \pm 0.07$ | $2.31 \pm 0.13$ |

(2) Setting the number of outer Bayesian optimization steps as 40.

| SGD Iteration | 500 | 1,000 | 2,000 | 3,000 | 4,000 |
|---|---|---|---|---|---|
| Performance | $2.56 \pm 0.53$ | $2.34 \pm 0.45$ | $2.29 \pm 0.29$ | $2.27 \pm 0.30$ | $2.22 \pm 0.16$ |

(3) Setting the number of outer Bayesian optimization steps as 60.

| SGD Iteration | 1,000 | 2,000 | 3,000 | 4,000 | 6,000 |
|---|---|---|---|---|---|
| Performance | $2.57 \pm 0.27$ | $2.26 \pm 0.28$ | $2.23 \pm 0.17$ | $2.19 \pm 0.19$ | $2.20 \pm 0.10$ |

When utilizing the UCB acquisition function for the outer Bayesian optimization, while keeping the number of steps for the outer Bayesian optimization fixed, we gradually increase the number of iterations for the inner SGD optimization. We can observe that initially, as the number of SGD iterations increases, the loss decreases gradually. However, when the number of SGD steps becomes excessively large, the decrease in loss tends to plateau.

## 6 CONCLUSION

Through solid analysis, we derive sublinear regret bounds guaranteeing simultaneous convergence of model parameters and hyperparameters to optimal configurations for generalization capability and providing insights into optimally balancing computations between model training and hyperparameter tuning. This theoretical foundation demonstrates the potential of Bayesian bilevel optimization for non-convex, derivative-free hyperparameter tuning. Future work may enrich this area through empirical studies, alternate acquisitions, and extensions to neural networks.

ACKNOWLEDGEMENT

We appreciate Hang Yu from the University of Sydney for his assistance in experiments. This work is supported by NSFC project No. 62222117 and the Fundamental Research Funds for the Central Universities under contracts WK3490000005 and KY2100000117.

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

## A   BAYESIAN BILEVEL OPTIMIZATION ALGORITHMS

In this section, we will present the algorithms employed in this study. To commence, we will introduce Bayesian bilevel optimization utilizing the expected improvement acquisition function as follows.

---

**Algorithm 1:** Bayesian bilevel Optimization with EI acquisition

---

**Input:** Place a Gaussian process prior on $L(\lambda, \theta_\lambda^*)$

1 **for** $t = 0, 1, ..., T - 1$ **do**

2      **for** $j = 0, 1, ..., N - 1$ **do**

3          Update model parameters using SGD:    $\theta_{\lambda_t}^{j+1} = \theta_{\lambda_t}^j - \eta_j \nabla_\theta \ell(\lambda_t, \theta_{\lambda_t}^j, z_{j_t}^{tr})$

4      **end**

5      Observe $y_t = L^{val}(\lambda_t, \theta_{\lambda_t}^N, S^{val})$ and Update posterior distribution on $L$ using all data.

6      Select the next hyperparameter $\lambda_{t+1}$ by maximizing the acquisition function $EI_t(\lambda)$ with the incumbent $\mu_t^+ = \min_{\lambda_i \in A_t} \mu_{t-1}(\lambda_i)$.

7
$$\lambda_{t+1} = \mathrm{argmax}_\lambda \, \mathbb{E}\left[\max\left(\mu_t^+ - L\left(\lambda, \theta_\lambda^*\right), 0\right) \mid H_{1:t}\right]$$

8 **end**

9 Return points $\lambda_t^+ = \mathrm{argmin}_{1 \leq i \leq t} \mu_{t-1}(\lambda_i)$ and $\theta_{\lambda_t^+}^N$ for all $1 \leq t \leq T$.

---

Next, we will introduce Bayesian bilevel optimization using the upper confidence bound acquisition function.

---

**Algorithm 2:** Bayesian bilevel optimization with UCB acquisition

---

**Input:** Place a Gaussian process prior on $L(\lambda, \theta_\lambda^*)$

1 **for** $t = 0, 1, ..., T - 1$ **do**

2      **for** $j = 0, 1, ..., N - 1$ **do**

3
$$\theta_{\lambda_t}^{j+1} = \theta_{\lambda_t}^j - \eta_j \nabla_\theta \ell\left(\lambda_t, \theta_{\lambda_t}^j, z_{j_t}^{tr}\right).$$

4      **end**

5      $y_t$ represents the observation of $L(\lambda_t, \theta_{\lambda_t}^*)$, where $y_t = L^{val}(\lambda_t, \theta_{\lambda_t}^N, S^{val})$.

6      Update the posterior probability distribution on $L$ using all available data.

7      Select the next hyperparameter $\lambda_{t+1}^+$ by maximizing the acquisition function $UCB_t(\lambda)$.

$$\lambda_{t+1} = \mathrm{argmax}_\lambda \left(-\mu_t(\lambda) + \beta_{t+1}\sigma_t(\lambda)\right)$$

8 **end**

9 Return points $\lambda_t^+ = \lambda_t$ and $\theta_{\lambda_t^+}^N$ for all $1 \leq t \leq T$.

---

## B   PROOF OF THEOREM 1

At the inner level, we keep the hyperparameters $\lambda$ fixed and treat them as constants, while we employ SGD to train and optimize the model parameters $\theta$. To establish an upper bound on the excess risk $L^{val}(\lambda, \theta_\lambda^N, S^{val}) - L(\lambda, \theta_\lambda^*)$, we introduce an effective decomposition method:

$$L^{val}\left(\lambda, \theta_\lambda^N, S^{val}\right) - L\left(\lambda, \theta_\lambda^*\right) = \underbrace{L^{val}\left(\lambda, \theta_\lambda^N, S^{val}\right) - L^{tr}\left(\lambda, \theta_\lambda^N, S^{tr}\right)}_{\text{Term 1}}$$

$$+ \underbrace{L^{tr}\left(\lambda, \theta_\lambda^N, S^{tr}\right) - L^{tr}\left(\lambda, \theta_\lambda^*, S^{tr}\right)}_{\text{Term 2}} + \underbrace{L^{tr}\left(\lambda, \theta_\lambda^*, S^{tr}\right) - L\left(\lambda, \theta_\lambda^*\right)}_{\text{Term 3}}. \quad \text{(B.1)}$$

Next, we proceed to upper bound the excess risk by individually upper bounding Term 1, Term 2, and Term 3.

**Upper Bounding Term 1**

Directly upper bounding Term 1 is challenging because there is no direct connection between the validation error $L^{val}\left(\lambda, \theta_\lambda^N, S^{val}\right)$ and the training error $L^{tr}\left(\lambda, \theta_\lambda^N, S^{tr}\right)$. However, we can

observe that both the validation set $S^{val}$ and the training set $S^{tr}$ are drawn from the distribution $\mathcal{D}$. Therefore, by using the expected error $L(\lambda, \theta_\lambda^N) = \mathbb{E}_{z \sim \mathcal{D}}[\ell(\lambda, \theta_\lambda^N, z)]$ as a bridge, we can establish a connection between the validation error and training error. This allows us to decompose $L^{val}\left(\lambda, \theta_\lambda^N, S^{val}\right) - L^{tr}\left(\lambda, \theta_\lambda^N, S^{tr}\right)$ into:

$$L^{val}\left(\lambda, \theta_\lambda^N, S^{val}\right) - L^{tr}\left(\lambda, \theta_\lambda^N, S^{tr}\right)$$
$$= \underbrace{L^{val}\left(\lambda, \theta_\lambda^N, S^{val}\right) - L(\lambda, \theta_\lambda^N)}_{\text{Term 4}} + \underbrace{L(\lambda, \theta_\lambda^N) - L^{tr}\left(\lambda, \theta_\lambda^N, S^{tr}\right)}_{\text{Term 5}}. \tag{B.2}$$

**Lemma 5.** *Let the validation error be $L^{val}(\lambda, \theta_\lambda^N, S^{val}) = \frac{1}{m}\sum_{i=1}^m \ell(\lambda, \theta_\lambda^N, z_i^{val})$. Similarly, let the training error be $L^{tr}(\lambda, \theta_\lambda^N, S^{tr}) = \frac{1}{n}\sum_{i=1}^n \ell(\lambda, \theta_\lambda^N, z_i^{tr})$. By choosing $N \asymp n \asymp m$ and $\eta \asymp N^{-\frac{1}{2}}$, we can establish that, with a probability of at least $1 - \delta$,*

$$L^{val}\left(\lambda, \theta_\lambda^N, S^{val}\right) - L^{tr}\left(\lambda, \theta_\lambda^N, S^{tr}\right) = \mathcal{O}\left(N^{-\frac{1}{2}}\log N\right).$$

To prove Lemma 5, we provide the definition of stability below, along with some useful concentration inequalities.

We use $\theta(S)$ to represent the model obtained when applying an algorithm $A$ to the dataset $S$. Note that we omit the dependency of the notation on $A$, which should be clear from the context. Additionally, we omit the hyperparameters $\lambda$ since we keep them fixed and treat them as constants at the inner level. Two datasets are considered neighboring datasets if they differ by only one example.

**Definition 3.** (Uniform Stability Bousquet & Elisseeff (2002)) A randomized algorithm $A$ is $\epsilon$ uniformly stable if for all neighboring datasets $S, S' \in \mathcal{D}$ we have $\sup_z \left[\ell\left(\theta(S), z\right) - \ell\left(\theta(S'), z\right)\right] \leq \epsilon$.

**Lemma 6** (McDiarmid's Inequality). *Consider independent random variables $Z_1, \cdots, Z_n \in \mathcal{Z}$ and a mapping $\phi : \mathcal{Z}^n \to \mathbb{R}$. If, for all $i \in \{1, \cdots, n\}$, and for all $z_1, \cdots, z_n, z_i' \in \mathcal{Z}$, the function $\phi$ satisfies*

$$\left|\phi\left(z_1, \cdots, z_{i-1}, z_i, z_{i+1}, \cdots, z_n\right) - \phi\left(z_1, \cdots, z_{i-1}, z_i', z_{i+1}, \cdots, z_n\right)\right| \leq c,$$

*then,*

$$P\left(\left|\phi\left(Z_1, \cdots, Z_n\right) - \mathbb{E}\phi\left(Z_1, \ldots, Z_n\right) \geq t\right|\right) \leq 2\exp\left(\frac{-2t^2}{nc^2}\right).$$

*Furthermore, for any $\delta \in (0, 1)$ the following inequality holds with probability at least $1 - \delta$*

$$\left|\phi\left(Z_1, \ldots, Z_n\right) - \mathbb{E}\left[\phi\left(Z_1, \ldots, Z_n\right)\right]\right| \leq \frac{c\sqrt{n\log(2/\delta)}}{\sqrt{2}}.$$

**Lemma 7** (Chernoff's Bound (Wainwright (2019))). *Let $X_1, \ldots, X_t$ be independent random variables taking values in $\{0, 1\}$. Let $X = \sum_{j=1}^t X_j$ and $\mu = \mathbb{E}[X]$. Then for any $\tilde{\delta} > 0$ with probability at least $1 - \exp\left(-\mu\tilde{\delta}^2/(2 + \tilde{\delta})\right)$ we have $X \leq (1 + \tilde{\delta})\mu$. Furthermore, for any $\delta \in (0, 1)$ with probability at least $1 - \delta$ we have*

$$X \leq \mu + \log(1/\delta) + \sqrt{2\mu\log(1/\delta)}.$$

**Lemma 8** (Generalization Error (Bousquet et al. (2020))). *We employ $\theta(S)$ to represent the model obtained when applying algorithm $A$ to the dataset $S$. Under the uniform stability condition (3) with parameter $\epsilon$ and the uniform boundedness of the loss function $\ell(\cdot, \cdot) \leq M$, we have that for any $\delta \in (0, 1)$, with a probability of at least $1 - \delta$:*

$$\left|L\left(\theta(S)\right) - L^{tr}\left(\theta(S)\right)\right| = \mathcal{O}\left(\epsilon\log n\log\left(\frac{1}{\delta}\right) + M\sqrt{n^{-1}\log\left(\frac{1}{\delta}\right)}\right).$$

*Proof of lemma 5.* For the sake of brevity in the proof, we omit the hyperparameters $\lambda$. Consider two samples, $S$ and $S'$, each consisting of $n$ data points, differing by only a single example. As we run SGD on these samples, we generate gradient updates $\theta_1, \ldots, \theta_N$ and $\theta_1', \ldots, \theta_N'$ for samples $S$ and

$S'$, respectively. Now, let's assume, without loss of generality, that the difference between $S$ and $S'$ is solely attributed to the first example. Applying the Lipschitz condition to $\ell(\cdot, z)$, we obtain:

$$|\ell(\theta_N, z) - \ell(\theta'_N, z)| \leq K\delta_N, \tag{B.3}$$

where $\delta_N = \|\theta_N - \theta'_N\|$. It's worth noting that during the $j$-th iteration, there is a probability of $(n-1)/n$ that the sample selected by SGD is the same in both sets $S$ and $S'$. The properties of convexity and $\gamma$-smoothness, as outlined in (Hardt et al. (2016)), establish that for any pair of $\theta$ and $\theta'$, the following holds:

$$\langle \nabla\ell(\theta, z) - \nabla\ell(\theta', z), \theta - \theta' \rangle \geq \frac{1}{\gamma}\|\nabla\ell(\theta, z) - \nabla\ell(\theta', z)\|^2.$$

In the event that $j_t$ is not equal to 1, based on the inequality $\eta_j \leq 2/\gamma$ we can deduce that:

$$\begin{aligned}
&\left\|\theta_{j+1} - \theta'_{j+1}\right\|^2 \\
&= \left\|\theta_j - \theta'_j\right\|^2 - 2\eta_j \left\langle \nabla\ell(\theta_j, z_{j_t}) - \nabla\ell(\theta'_j, z'_{j_t}), \theta_j - \theta'_j \right\rangle + \eta_j^2 \left\|\nabla\ell(\theta_j, z_{j_t}) - \nabla\ell(\theta'_j, z'_{j_t})\right\|^2 \\
&\leq \left\|\theta_j - \theta'_j\right\|^2 - \left(\frac{2\eta_j}{\gamma} - \eta_j^2\right) \left\|\nabla\ell(\theta_j, z_{j_t}) - \nabla\ell(\theta'_j, z'_{j_t})\right\|^2 \\
&\leq \left\|\theta_j - \theta'_j\right\|^2. \tag{B.4}
\end{aligned}$$

In the scenario where the probability of selecting a different example is $1/n$, i.e., $j_t = 1$, we can utilize the triangle inequality and the previously mentioned inequality to establish the following:

$$\begin{aligned}
&\left\|\theta_{j+1} - \theta'_{j+1}\right\| \\
&= \left\|\theta_j - \eta_j \nabla\ell(\theta_j, z_{j_t}) - (\theta'_j - \eta_j \nabla\ell(\theta'_j, z_{j_t}))\right\| + \eta_j \|\nabla\ell(\theta'_j, z'_{j_t}) - \nabla\ell(\theta'_j, z_{j_t})\| \\
&\leq \left\|\theta_j - \theta'_j\right\| + 2\eta_j K.
\end{aligned}$$

By merging the two cases mentioned above, we can conclude that for every $j$:

$$\left\|\theta_{j+1} - \theta'_{j+1}\right\| \leq \left\|\theta_j - \theta'_j\right\| + 2\eta_j K \mathbb{I}_{[j_t=1]},$$

where $\mathbb{I}$ represents the indicator function. By solving the recursive inequality, we obtain:

$$\left\|\theta_{j+1} - \theta'_{j+1}\right\| \leq 2K \sum_{j=1}^{N} \eta_j \mathbb{I}_{[j_t=1]} \leq 2K\eta \sum_{j=1}^{N} \mathbb{I}_{[j_t=1]}. \tag{B.5}$$

Invoking Lemma 7 whereby $X_j = \mathbb{I}_{[j_t=1]}$ and $\mu = N/n$, we obtain the subsequent inequality that holds with a probability of at least $1 - \delta$:

$$\sum_{j=1}^{N} \mathbb{I}_{[j_t=1]} \leq N/n + \log(1/\delta) + \sqrt{2Nn^{-1}\log(1/\delta)}.$$

Consequently, with a probability at least $1 - \delta$, the following inequality is satisfied:

$$\left\|\theta_{j+1} - \theta'_{j+1}\right\| \leq 2K\eta(N/n + \log(1/\delta) + \sqrt{2Nn^{-1}\log(1/\delta)}).$$

Then we derive the ensuing inequality with a probability of at least $1 - \delta$:

$$\|\theta(S) - \theta(S')\| \leq 2K\eta(N/n + \log(1/\delta) + \sqrt{2Nn^{-1}\log(1/\delta)}).$$

Substituting the inequality back into Eq. B.3, we arrive at the following result holding with a probability of at least $1 - \delta$:

$$|\ell(\theta(S); z) - \ell(\theta(S'); z)| \leq 2K^2\eta(N/n + \log(1/\delta) + \sqrt{2Nn^{-1}\log(1/\delta)}). \tag{B.6}$$

Subsequently, we can bound Term 5 by applying lemma 8 with a stability parameter $\epsilon = 2K^2\eta(N/n + \log(1/\delta) + \sqrt{2Nn^{-1}\log(1/\delta)})$. With a probability of at least $1 - \delta$, we obtain:

$$L\left(\lambda, \theta_\lambda^N\right) - L^{tr}\left(\lambda, \theta_\lambda^N, S^{tr}\right) = \mathcal{O}\left(\eta Nn^{-1}\log n \log(1/\delta) + M\sqrt{n^{-1}\log(1/\delta)}\right). \tag{B.7}$$

Next, we begin by addressing Term 4, denoted as $L^{val}(\lambda, \theta_\lambda^N, S^{val}) - L(\lambda, \theta_\lambda^N)$, where $S^{val} = \{z_i^{val}\}_{i=1}^m$. Notably, $\theta_\lambda^N$ is obtained by training on $S^{tr}$, which is independent of $S^{val}$. Then, we have:

$$L\left(\lambda, \theta_\lambda^N\right) = \mathbb{E}[L^{\mathrm{val}}\left(\lambda, \theta_\lambda^N, S^{\mathrm{val}}\right)].$$

Furthermore, since $\ell(\lambda, \theta, z)$ is uniformly bounded by $M$, we can observe that for $S^{val}$ and $S^{val'}$ differing by only one example, we have

$$L^{val}\left(\lambda, \theta_\lambda^N, S^{val}\right) - L^{val}\left(\lambda, \theta_\lambda^N, S^{val'}\right) = \frac{1}{m}\left(\sum_{i=1}^m \ell\left(\lambda, \theta_\lambda^N, z_i^{val}\right) - \sum_{i=1}^m \ell\left(\lambda, \theta_\lambda^N, z_i^{val'}\right)\right) \leq \frac{M}{m}.$$

Then we can apply lemma 6 with $c = \frac{M}{m}$ to obtain, with probability at least $1 - \delta$,

$$|L^{\mathrm{val}}\left(\lambda, \theta_\lambda^N, S^{\mathrm{val}}\right) - L\left(\lambda, \theta_\lambda^N\right)| \leq M\sqrt{\frac{1}{2m}\log(2/\delta)}. \tag{B.8}$$

By substituting inequalities B.8 and B.7 into equation B.2, we can establish that, with a probability of at least $1 - \delta$,

$$L^{val}\left(\lambda, \theta_\lambda^N, S^{val}\right) - L^{tr}\left(\lambda, \theta_\lambda^N, S^{tr}\right)$$
$$= \mathcal{O}\left(\eta N n^{-1} \log n \log(1/\delta) + M\sqrt{n^{-1}\log(1/\delta)} + M\sqrt{m^{-1}\log(2/\delta)}\right).$$

By the choice of $N \asymp n \asymp m$ and $\eta \asymp N^{-\frac{1}{2}}$, we can deduce that, with a probability of at least $1 - \delta$,

$$L^{val}\left(\lambda, \theta_\lambda^N, S^{val}\right) - L^{tr}\left(\lambda, \theta_\lambda^N, S^{tr}\right) = \mathcal{O}\left(N^{-\frac{1}{2}}\log N\right). \tag{B.9}$$

$\square$

**Upper Bounding Term 2**

We introduce the following lemma to bound Term 2, which quantifies the optimization error.

**Lemma 9** (Optimization Error (Lei & Tang, 2018))**.** *Assuming that the function $\ell(\theta)$ is convex and $L$-Lipschitz continuous with respect to $\theta$, we run SGD with step sizes $\eta_t = \eta \asymp \frac{1}{\sqrt{N}}$. Then, with a probability of at least $1 - \delta$, we can establish that:*

$$L^{tr}(\theta) - \inf_\theta L^{tr}(\theta) = \mathcal{O}(N^{-\frac{1}{2}}\log^{\frac{3}{2}}(N/\delta)).$$

**Lemma 10.** *Select $\delta \in (0, 1)$. Set the step size of SGD as $\eta_t = \eta \asymp \frac{1}{\sqrt{N}}$. Then, with a probability of $1 - \delta$, we achieve:*

$$L^{tr}\left(\lambda, \theta_\lambda^N, S^{tr}\right) - L^{tr}\left(\lambda, \theta_\lambda^*, S^{tr}\right) = \mathcal{O}\left(N^{-\frac{1}{2}}\log^{\frac{3}{2}}(N/\delta)\right).$$

*Proof of Lemma 10.* We should take note that, based on equality 3.1, with $\theta_\lambda^* = \arg\min_{\theta \in \Theta} L(\lambda, \theta)$, we can derive

$$L^{tr}\left(\lambda, \theta_\lambda^*, S^{tr}\right) \geq \inf_\theta L^{tr}(\lambda, \theta, S^{tr}).$$

From lemma 9, we can obtain

$$L^{tr}\left(\lambda, \theta_\lambda^N, S^{tr}\right) - \inf_\theta L^{tr}(\lambda, \theta, S^{tr}) = \mathcal{O}\left(N^{-\frac{1}{2}}\log^{\frac{3}{2}}(N/\delta)\right).$$

Combining the above two inequalities, we derive

$$L^{tr}\left(\lambda, \theta_\lambda^N, S^{tr}\right) - L^{tr}\left(\lambda, \theta_\lambda^*, S^{tr}\right)$$
$$= L^{tr}\left(\lambda, \theta_\lambda^N, S^{tr}\right) - \inf_\theta L^{tr}\left(\lambda, \theta, S^{tr}\right) + \inf_\theta L^{tr}\left(\lambda, \theta, S^{tr}\right) - L^{tr}\left(\lambda, \theta_\lambda^*, S^{tr}\right)$$
$$= \mathcal{O}\left(N^{-\frac{1}{2}}\log^{\frac{3}{2}}(N/\delta)\right).$$

$\square$

**Upper Bounding Term 3**

**Lemma 11.** *Choose a value for $\delta$ from the interval $(0,1)$. Then, with a probability of $1 - \delta$, we obtain:*

$$L^{tr}\left(\lambda, \theta_\lambda^*, S^{tr}\right) - L\left(\lambda, \theta_\lambda^*\right) = \mathcal{O}\left(N^{-\frac{1}{2}}\right).$$

*Proof of lemma 11.* It is worthy of note that $\theta_\lambda^*$ is independent of $S^{tr}$. Moreover, as $\ell(\lambda, \theta, z)$ is uniformly bounded by M, we can discern that for $S^{tr}$ and $S^{tr'}$ differing by only one example, there exists:

$$L^{tr}\left(\lambda, \theta_\lambda^*, S^{tr}\right) - L^{tr}\left(\lambda, \theta_\lambda^*, S^{tr'}\right) = \frac{1}{n}\left(\sum_{i=1}^{n} \ell\left(\lambda, \theta_\lambda^*, z_i^{tr}\right) - \sum_{i=1}^{n} \ell\left(\lambda, \theta_\lambda^*, z_i^{tr'}\right)\right) \leq \frac{M}{n}.$$

Consequently, we can invoke Lemma 6 with $c = \frac{M}{n}$ to attain, with a probability of at least $1 - \delta$:

$$L^{tr}\left(\lambda, \theta_\lambda^*, S^{tr}\right) - L\left(\lambda, \theta_\lambda^*\right) \leq M\sqrt{\frac{1}{2n}\log(2/\delta)}.$$

By the choice of $N \asymp n$, we can deduce that, with a probability of at least $1 - \delta$,

$$L^{tr}\left(\lambda, \theta_\lambda^*, S^{tr}\right) - L\left(\lambda, \theta_\lambda^*\right) = \mathcal{O}\left(N^{-\frac{1}{2}}\right).$$

$\square$

Through the aforementioned handling of Terms 1, 2, and 3, we can ultimately derive the excess risk bound that we are interested in.

*Proof of Theorem 1.* Combining Term 1 in lemma 5, Term 2 in lemma 10 and Term 3 in lemma 11 with the decomposition B.1, by the choice of $N \asymp n \asymp m$ and $\eta \asymp N^{-\frac{1}{2}}$, we can deduce that, with a probability of at least $1 - \delta$,

$$L^{\text{val}}\left(\lambda, \theta_\lambda^N, S^{\text{val}}\right) - L\left(\lambda, \theta_\lambda^*\right) =$$

$$\mathcal{O}\left(\left(K^2 \log N \log(1/\delta) + M\sqrt{\log(1/\delta)} + M\sqrt{\log(2/\delta)} + \log^{3/2}(N/\delta)\right)\left(N^{-1/2}\right)\right)$$

When we neglect constants like $K, M$, and $\delta$, we can derive the result:

$$L^{val}\left(\lambda, \theta_\lambda^N, S^{val}\right) - L\left(\lambda, \theta_\lambda^*\right) = \mathcal{O}\left(N^{-\frac{1}{2}} \log^{\frac{3}{2}} N\right).$$

The proof is complete. $\square$

## C   PROOF OF LEMMA 2

In previous works, the assumptions regarding the noise $\varepsilon_t$ differ from those in our work. Srinivas et al. (2010) and Nguyen & Gupta (2017) assume that the noise is uniformly bounded by $\sigma$, whereas Chowdhury & Gopalan (2017) and Gupta et al. (2022) assume that the noise is conditionally $R$-sub-Gaussian. Furthermore, most prior research assumes that the noise sequence $\{\varepsilon_t\}_{t=1}^{T}$ is a martingale difference sequence. In contrast, our noise follows $\varepsilon_t = L^{val}\left(\lambda_t, \theta_{\lambda_t}^N, S^{val}\right) - L\left(\lambda_t, \theta_{\lambda_t}^*\right)$, which clearly does not align with the assumptions made in previous works. Consequently, we cannot directly apply theorems similar to those in previous research that express the concentration of the posterior mean around the true objective function.

We follow the framework in Srinivas et al. (2010) to prove lemma 2.

*Proof of lemma 2.* Let's revisit the posterior mean function denoted as $\mu_t(\cdot)$ and the posterior covariance function denoted as $k_t(\cdot, \cdot)$ as discussed in Section 3.2.2, taking into account the data points $(\lambda_i, y_i)$ for $i = 1, \ldots, t$. It becomes apparent that the RKHS norm associated with the kernel $k_t$ can be expressed as:

$$\|L(\cdot, \theta_\cdot^*)\|_{k_t}^2 = \|L(\cdot, \theta_\cdot^*)\|_k^2 + \sigma^{-2} \sum_{i=1}^{t} L\left(\lambda_i, \theta_{\lambda_i}^*\right)^2. \tag{C.1}$$

This suggests that $\mathcal{H}_k(\Lambda)$ and $\mathcal{H}_{k_t}(\Lambda)$ are equivalent for any $t$. Given that the reproducing property holds, i.e., $\langle L(\cdot, \theta^*_\cdot), k_t(\cdot, \lambda) \rangle_{k_t} = L(\lambda, \theta^*_\lambda)$ for any $L(\cdot, \theta^*_\cdot) \in \mathcal{H}_{k_t}(\Lambda)$, then,

$$
\begin{aligned}
|\mu_t(\lambda) - L(\lambda, \theta^*_\lambda)| &\leq k_t(\lambda, \lambda)^{1/2} \|\mu_t - L(\cdot, \theta^*_\cdot)\|_{k_t} \\
&= \sigma_t(\lambda) \|\mu_t - L(\cdot, \theta^*_\cdot)\|_{k_t}.
\end{aligned}
\tag{C.2}
$$

Recall that the posterior mean function is given by:

$$
\mu_t(\lambda) = \boldsymbol{k}_t(\lambda)^T \left( \boldsymbol{K}_t + \sigma^2 \boldsymbol{I} \right)^{-1} \boldsymbol{y}_t.
$$

Then, we also have:

$$
\langle \mu_t, L(\cdot, \theta^*_\cdot) \rangle_k = \boldsymbol{L}_t(\lambda, \theta^*_\lambda)^T \left( \boldsymbol{K}_t + \sigma^2 \boldsymbol{I} \right)^{-1} \boldsymbol{y}_t,
\tag{C.3}
$$

where $\boldsymbol{L}_t(\lambda, \theta^*_\lambda) = [L(\lambda_1, \theta^*_{\lambda_1}), \ldots, L(\lambda_t, \theta^*_{\lambda_t})]^T$. Additionally, the RKHS norm of $\mu_t$ is given by:

$$
\|\mu_t\|^2_k = \boldsymbol{y}_t^T \left( \boldsymbol{K}_t + \sigma^2 \boldsymbol{I} \right)^{-1} \boldsymbol{y}_t - \sigma^2 \| \left( \boldsymbol{K}_t + \sigma^2 \boldsymbol{I} \right)^{-1} \boldsymbol{y}_t \|^2.
\tag{C.4}
$$

Moreover, for $i \leq t$, we obtain:

$$
\begin{aligned}
\mu_t(\lambda_i) &= \boldsymbol{\delta}_{i,t}^T \boldsymbol{K}_t \left( \boldsymbol{K}_t + \sigma^2 \boldsymbol{I} \right)^{-1} \boldsymbol{y}_t \\
&= \boldsymbol{\delta}_{i,t}^T (\boldsymbol{K}_t + \sigma^2 \boldsymbol{I}) \left( \boldsymbol{K}_t + \sigma^2 \boldsymbol{I} \right)^{-1} \boldsymbol{y}_t - \sigma^2 \boldsymbol{\delta}_{i,t}^T \left( \boldsymbol{K}_t + \sigma^2 \boldsymbol{I} \right)^{-1} \boldsymbol{y}_t \\
&= y_i - \sigma^2 \boldsymbol{\delta}_{i,t}^T \left( \boldsymbol{K}_t + \sigma^2 \boldsymbol{I} \right)^{-1} \boldsymbol{y}_t.
\end{aligned}
\tag{C.5}
$$

where $\boldsymbol{\delta}_{i,t}$ represents a $t$-dimensional vector where only the $i$-th element is equal to 1, while all other elements are set to 0. Then, referring to Equation C.1, we can derive:

$$
\|\mu_t - L(\cdot, \theta^*_\cdot)\|^2_{k_t} = \|\mu_t - L(\cdot, \theta^*_\cdot)\|^2_k + \sigma^{-2} \sum_{i=1}^t (\mu_t(\lambda_i) - L(\lambda_i, \theta^*_{\lambda_i}))^2.
$$

$$
= \|\mu_t\|^2_k + \|L(\cdot, \theta^*_\cdot)\|^2_k - 2 \langle \mu_t, L(\cdot, \theta^*_\cdot) \rangle_k + \sigma^{-2} \sum_{i=1}^t (\mu_t(\lambda_i) - L(\lambda_i, \theta^*_{\lambda_i}))^2.
\tag{C.6}
$$

The noise sequence is $\boldsymbol{\varepsilon}_t = \{\varepsilon_i\}_{i=1}^t$ with $\varepsilon_i = y_i - L(\lambda_i, \theta^*_{\lambda_i})$. Taking into account equation C.5, we can express this as follows:

$$
\begin{aligned}
\sigma^{-2} \sum_{i=1}^t (\mu_t(\lambda_i) - L(\lambda_i, \theta^*_{\lambda_i}))^2 &= \sigma^{-2} \sum_{i=1}^t (\varepsilon_i - \sigma^2 \boldsymbol{\delta}_{i,t}^T \left( \boldsymbol{K}_t + \sigma^2 \boldsymbol{I} \right)^{-1} \boldsymbol{y}_t)^2 \\
&\leq \sigma^2 \| \left( \boldsymbol{K}_t + \sigma^2 \boldsymbol{I} \right)^{-1} \boldsymbol{y}_t \|^2 - 2 \boldsymbol{\varepsilon}_t^T \left( \boldsymbol{K}_t + \sigma^2 \boldsymbol{I} \right)^{-1} \boldsymbol{y}_t + \sigma^{-2} \|\boldsymbol{\varepsilon}_t\|^2
\end{aligned}
\tag{C.7}
$$

Considering $\|L(\cdot, \theta^*_\cdot)\|_k$ is bounded by $B$, the upper bound of the noise from Theorem 1 and the positive definiteness of $\boldsymbol{K}_t + \sigma^2 \boldsymbol{I}$, we can substitute equations C.3, C.4, and C.7 into equation C.6 to derive, with a probability at least $1 - \delta$:

$$
\begin{aligned}
\|\mu_t - L(\cdot, \theta^*_\cdot)\|^2_{k_t} &= \|L(\cdot, \theta^*_\cdot)\|^2_k - \boldsymbol{y}_t^T \left( \boldsymbol{K}_t + \sigma^2 \boldsymbol{I} \right)^{-1} \boldsymbol{y}_t + \sigma^{-2} \|\boldsymbol{\varepsilon}_t\|^2 \\
&\leq B^2 + \sigma^{-2} t \varphi^2(N) N^{-1}.
\end{aligned}
\tag{C.8}
$$

Combining inequalities C.2 and C.8, we obtain that

$$
|\mu_t(\lambda) - L(\lambda, \theta^*_\lambda)| \leq \sqrt{B^2 + \sigma^{-2} t \varphi^2(N) N^{-1}} \, \sigma_t(\lambda).
$$

The proof is completed. $\qquad\square$

# D  PROOF OF THEOREM 3

First, we will bound the instantaneous regret, given by $r_t = L\left(\lambda_t^+, \theta_{\lambda_t^+}^N\right) - L\left(\lambda^*, \theta_{\lambda^*}^*\right)$. Next, we will aim to find an upper bound for the sum $R_T = \sum_{t=1}^T r_t$.

To derive an upper bound for $r_t$, we decompose it into three constituent terms as follows:

$$
\begin{aligned}
r_t &= L\left(\lambda_t^+, \theta_{\lambda_t^+}^N\right) - L\left(\lambda^*, \theta_{\lambda^*}^*\right) \\
&= \underbrace{L\left(\lambda_t^+, \theta_{\lambda_t^+}^N\right) - L\left(\lambda_t^+, \theta_{\lambda_t^+}^*\right)}_{\text{Term 6}} + \underbrace{L\left(\lambda_t^+, \theta_{\lambda_t^+}^*\right) - \mu_t^+}_{\text{Term 7}} + \underbrace{\mu_t^+ - L\left(\lambda^*, \theta_{\lambda^*}^*\right)}_{\text{Term 8}}.
\end{aligned}
\tag{D.1}
$$

We introduce the function $\tau(z) = z\Phi(z) + \phi(z)$, where $\Phi$ and $\phi$ represent the cumulative distribution function (c.d.f.) and probability density function (p.d.f.) of the standard normal distribution, respectively. Now, we define the function $I_t(\lambda)$ as follows: $I_t(\lambda) = \max\left\{\mu_t^+ - L\left(\lambda, \theta_\lambda^*\right), 0\right\}$. In this expression, $I_t(\lambda)$ assumes positive values when the prediction is lower than the minimum GP mean observed so far up to that point. For all other cases, $I_t(\lambda)$ is set to zero. The process of finding the new query point involves maximizing the expected improvement:

$$
\lambda_t = \operatorname{argmax}_\lambda \mathbb{E}\left(I_t(\lambda)\right).
$$

**Upper Bounding Term 6.**

**Lemma 12.** *Choose $\delta \in (0,1)$ and $N \asymp m$. Then with probability at least $1 - \delta$, we have*

$$
L\left(\lambda_t^+, \theta_{\lambda_t^+}^N\right) - L\left(\lambda_t^+, \theta_{\lambda_t^+}^*\right) \leq \alpha(N)N^{-\frac12},
$$

*where $\alpha(N) = \mathcal{O}\left(\log^{\frac32} N\right)$.*

*Proof of lemma 12.* We decompose Term 6 into two constituent terms.

$$
\begin{aligned}
&L\left(\lambda_t^+, \theta_{\lambda_t^+}^N\right) - L\left(\lambda_t^+, \theta_{\lambda_t^+}^*\right) \\
&= L\left(\lambda_t^+, \theta_{\lambda_t^+}^N\right) - L^{val}(\lambda_t^+, \theta_{\lambda_t^+}^N, S^{val}) + L^{val}(\lambda_t^+, \theta_{\lambda_t^+}^N, S^{val}) - L\left(\lambda_t^+, \theta_{\lambda_t^+}^*\right).
\end{aligned}
$$

In regard to $L\left(\lambda_t^+, \theta_{\lambda_t^+}^N\right) - L^{val}\left(\lambda_t^+, \theta_{\lambda_t^+}^N, S^{val}\right)$, it is noteworthy that $\theta_{\lambda_t^+}^N$ is derived through training on the dataset $S^{tr}$, which remains independent of the dataset $S^{val}$. Consequently, we can establish the following relationship:

$$
L\left(\lambda_t^+, \theta_{\lambda_t^+}^N\right) = \mathbb{E}\left[L^{val}\left(\lambda_t^+, \theta_{\lambda_t^+}^N, S^{val}\right)\right].
$$

Moreover, given that $\ell(\lambda, \theta, z)$ is uniformly bounded by $M$, when considering datasets $S^{val}$ and $S^{val'}$, which differ by only a single example, the following assertion can be made:

$$
\begin{aligned}
&L^{val}\left(\lambda_t^+, \theta_{\lambda_t^+}^N, S^{val}\right) - L^{val}\left(\lambda_t^+, \theta_{\lambda_t^+}^N, S^{val'}\right) \\
&= \frac1m\left(\sum_{i=1}^m \ell\left(\lambda_t^+, \theta_{\lambda_t^+}^N, z_i^{val}\right) - \sum_{i=1}^m \ell\left(\lambda_t^+, \theta_{\lambda_t^+}^N, z_i^{val'}\right)\right) \leq \frac{M}{m}.
\end{aligned}
$$

Then we can apply lemma 6 with $c = \frac{M}{m}$ to obtain, with probability at least $1 - \delta$,

$$
\left|L^{val}\left(\lambda_t^+, \theta_{\lambda_t^+}^N, S^{val}\right) - L\left(\lambda_t^+, \theta_{\lambda_t^+}^N\right)\right| \leq M\sqrt{\frac{1}{2m}\log(2/\delta)}.
$$

As for the second term $L^{val}\left(\lambda_t^+, \theta_{\lambda_t^+}^N, S^{val}\right) - L\left(\lambda_t^+, \theta_{\lambda_t^+}^*\right)$, we can apply the result from Theorem 1 to obtain:

$$
L^{val}\left(\lambda_t^+, \theta_{\lambda_t^+}^N, S^{val}\right) - L\left(\lambda_t^+, \theta_{\lambda_t^+}^*\right) \leq \varphi(N)N^{-\frac12},
$$

where $\varphi(N) = \mathcal{O}\left(\log^{\frac{3}{2}} N\right)$. Then, combining the above two inequalities and choosing $N \asymp m$, we can deduce that, with a probability of at least $1 - \delta$,

$$L\left(\lambda_t^+, \theta_{\lambda_t^+}^N\right) - L\left(\lambda_t^+, \theta_{\lambda_t^+}^*\right) \le \alpha(N)N^{-\frac{1}{2}}, \tag{D.2}$$

where $\alpha(N) = \mathcal{O}\left(\log^{\frac{3}{2}} N\right)$. The proof is complete. $\qquad\square$

**Upper Bounding Term 7.**

When addressing Terms 7 and 8, we adopt the proof framework outlined in Gupta et al. (2022), with the ability to adapt and apply certain lemmas from Gupta et al. (2022) to our proof following appropriate adjustments.

**Lemma 13.** *(Gupta et al. (2022)) Choose a value for $\delta$ from the interval $(0, 1)$. Then, with a probability of $1 - \delta$, we obtain*

$$L\left(\lambda_t^+, \theta_{\lambda_t^+}^*\right) - \mu_t^+ \le \beta_t \left(\sqrt{2\pi}\left(\beta_t + 1\right)\sigma_{t-1}\left(\lambda_t\right) + \sqrt{2\pi}I_t(\lambda_t)\right).$$

**Upper Bounding Term 8.**

Initially, we establish both the lower and upper bounds for the acquisition function. The following lemma draws inspiration from Lemma 9 in Wang & de Freitas (2014).

**Lemma 14.** *Choose $\delta \in (0, 1)$. For $\lambda \in \Lambda, t \in T$, define $I_t(\lambda) = \max\left\{0, \mu_t^+ - L\left(\lambda, \theta_\lambda^*\right)\right\}$. Then with probability at least $1 - \delta$ we have*

$$I_t(\lambda) - \beta_t\sigma_{t-1}(\lambda) \le EI_t(\lambda) \le I_t(\lambda) + \left(\beta_t + 1\right)\sigma_{t-1}(\lambda).$$

*Proof of lemma 14.* If $\sigma_{t-1}(\lambda) = 0$, then $EI_t(\lambda) = I_t(\lambda) = 0$, which leads to a trivial result. Now, let's assume that $\sigma_{t-1}(\lambda) > 0$. Define $q = \frac{\mu_t^+ - L(\lambda, \theta_\lambda^*)}{\sigma_{t-1}(\lambda)}$ and $u = \frac{\mu_t^+ - \mu_{t-1}(\lambda)}{\sigma_{t-1}(\lambda)}$. We can then establish that:

$$EI_t(\lambda) = \sigma_{t-1}(\lambda)\tau(u).$$

According to lemma 2, we can conclude that $|u - q| \le \beta_t$ with a probability of at least $1 - \delta$. Since $\tau'(z) > 0$, $\tau$ is a non-decreasing function, and for $z > 0$, $\tau(z) \le 1 + z$. Therefore, we have:

$$\begin{aligned}
EI_t(\lambda) &\le \sigma_{t-1}(\lambda)\tau\left(\max\{0, q\} + \beta_t\right) \\
&\le \sigma_{t-1}(\lambda)\left(\max\{0, q\} + \beta_t + 1\right) \\
&= I_t(\lambda) + \left(\beta_t + 1\right)\sigma_{t-1}(\lambda).
\end{aligned}$$

If $I_t(\lambda) = 0$, the lower bound is straightforward as $EI_t(\lambda)$ is non-negative. Let's consider the case where $I_t(\lambda) > 0$. Since $EI_t(\lambda) \ge 0$ and $\tau(z) \ge 0$ for all $z$, and $\tau(z) = z + \tau(-z) \ge z$. Therefore, we can write:

$$\begin{aligned}
EI_t(\lambda) &\ge \sigma_{t-1}(\lambda)\tau\left(q - \beta_t\right) \\
&\ge \sigma_{t-1}(\lambda)\left(q - \beta_t\right) \\
&= I_t(\lambda) - \beta_t\sigma_{t-1}(\lambda).
\end{aligned}$$

The proof is now complete. $\qquad\square$

**Lemma 15.** *Choose $\delta \in (0, 0.5)$. Then, with a probability of at least $1 - 2\delta$, we obtain:*

$$\mu_t^+ - L\left(\lambda^*, \theta_{\lambda^*}^*\right) \le \left(\frac{1 + \beta_t}{\tau(\beta_t) - \beta_t}\right)\left(I_t\left(\lambda_t\right) + \left(\beta_t + 1\right)\sigma_{t-1}\left(\lambda_t\right)\right).$$

*Proof of lemma 15.* If $\sigma_{t-1}(\lambda^*) = 0$, as per the definition of $EI_t(\lambda)$, we find that $EI_t(\lambda^*) = I_t(\lambda^*) = 0$, which leads to a trivial result. Now, let's consider the case where $\sigma_{t-1}(\lambda^*) > 0$. If $L\left(\lambda^*, \theta_{\lambda^*}^*\right) > \mu_t^+$, the lemma remains trivial. We will now explore the case where $L\left(\lambda^*, \theta_{\lambda^*}^*\right) \le \mu_t^+$.

Based on lemma 2, we know that $L(\lambda^*, \theta^*_{\lambda^*}) - \mu_{t-1}(\lambda^*) \geq -\beta_t \sigma_{t-1}(\lambda^*)$ with a probability of $1 - \delta$. Combining this with the fact that $L(\lambda^*, \theta^*_{\lambda^*}) \leq \mu_t^+$, we can conclude that with a probability of $1 - \delta$, the following holds:

$$\frac{\mu_t^+ - \mu_{t-1}(\lambda^*)}{\sigma_{t-1}(\lambda^*)} \geq -\beta_t.$$

Recalling the definition of the EI acquisition function, with a probability of at least $1 - \delta$, we can conclude that:

$$EI_t(\lambda^*) = \sigma_{t-1}(\lambda^*) \tau\left(\frac{\mu_t^+ - \mu_{t-1}(\lambda^*)}{\sigma_{t-1}(\lambda^*)}\right) \geq \sigma_{t-1}(\lambda^*) \tau(-\beta_t).$$

Combining the above inequality with the previously established result $EI_t(\lambda^*) \geq I_t(\lambda^*) - \beta_t \sigma_{t-1}(\lambda^*)$ as proven in Lemma 14, we obtain:

$$(\frac{\beta_t}{\tau(-\beta_t)} + 1) EI_t(\lambda^*) \geq I_t(\lambda^*).$$

Using the above inequality in conjunction with the fact that $\tau(z) = z + \tau(-z)$ for $z = \beta_t$, we obtain

$$I_t(\lambda^*) \leq \left(\frac{\tau(\beta_t)}{\tau(\beta_t) - \beta_t}\right) EI_t(\lambda^*) \leq \left(\frac{1 + \beta_t}{\tau(\beta_t) - \beta_t}\right) EI_t(\lambda^*). \tag{D.3}$$

In accordance with Algorithm 1, $\lambda_t = \arg\max EI_t(\lambda)$. When combined with the upper bound from Lemma 14, we can deduce:

$$EI_t(\lambda^*) \leq EI_t(\lambda_t) \leq I_t(\lambda_t) + (\beta_t + 1)\sigma_{t-1}(\lambda_t) \tag{D.4}$$

Utilizing the inequalities D.3 and D.4, we obtain

$$\mu_t^+ - L(\lambda^*, \theta^*_{\lambda^*}) \leq I_t(\lambda^*)$$
$$\leq \left(\frac{1 + \beta_t}{\tau(\beta_t) - \beta_t}\right) EI_t(\lambda^*)$$
$$\leq \left(\frac{1 + \beta_t}{\tau(\beta_t) - \beta_t}\right)(I_t(\lambda_t) + (\beta_t + 1)\sigma_{t-1}(\lambda_t)). \tag{D.5}$$

The proof of lemma 15 is complete. $\qquad\square$

**Upper bounding the immediate regret.** $r_t = L\left(\lambda_t^+, \theta^N_{\lambda_t^+}\right) - L(\lambda^*, \theta^*_{\lambda^*})$. Combining lemma 12 for Term 6, lemma 13 for Term 7 and lemma 15 for Term 8, we can conclude with high probability that:

$$r_t \leq \left(\frac{1 + \beta_t}{\tau(\beta_t) - \beta_t} + \sqrt{2\pi}\beta_t\right)[I_t(\lambda_t) + (\beta_t + 1)\sigma_{t-1}(\lambda_t)] + \alpha(N)N^{-\frac{1}{2}}. \tag{D.6}$$

**Upper bounding the cumulative regret.** $R_T = \sum_{t=1}^T r_t$. Based on the above inequality, we can derive an upper bound for the cumulative regret as follows:

$$\sum_{t=1}^T r_t \leq \left(\frac{1 + \beta_T}{\tau(\beta_T) - \beta_T} + \sqrt{2\pi}\beta_T\right)\left(\underbrace{\sum_{t=1}^T I_t(\lambda_t)}_{\text{Term 9}} + (\beta_T + 1)\underbrace{\sum_{t=1}^T \sigma_{t-1}(\lambda_t)}_{\text{Term 10}}\right) + T\alpha(N)N^{-\frac{1}{2}}.$$
$$\tag{D.7}$$

We now introduce lemma 16 and lemma 17 from Gupta et al. (2022) and Chowdhury & Gopalan (2017) to assist in bounding Term 9 and Term 10.

**Lemma 16.** *Gupta et al. (2022) Select $\delta$ from the interval $(0, 1)$. Then, with a probability of at least $1 - \delta$, we can establish that:*

$$\sum_{t=1}^T I_t(\lambda_t) = \mathcal{O}\left(\beta_T \sqrt{T\gamma_T}\right).$$

**Lemma 17.** *Chowdhury & Gopalan (2017) Let $\lambda_1, \ldots, \lambda_t$ represent the points selected by algorithm 1. The sum of the predictive standard deviations at these points can be expressed in terms of the maximum information gain. To be more precise:*

$$\sum_{t=1}^{T} \sigma_{t-1}(\lambda_t) = \mathcal{O}\left(\sqrt{T\gamma_T}\right).$$

Ultimately, based on the above analysis, we can demonstrate Theorem 3 as follows.

*Proof of Theorem 3.* By combining the results in Lemma 16 and Lemma 17 with Inequality D.7, we derive an upper bound for the cumulative regret $\sum_{t=1}^{T} r_t$ as follows:

$$R_T = \sum_{t=1}^{T} r_t = \mathcal{O}\left(\frac{\beta_T^2 \sqrt{T\gamma_T}}{\tau(\beta_T) - \beta_T} + TN^{-\frac{1}{2}}\right). \tag{D.8}$$

where $\beta_T = \sqrt{B^2 + \sigma^{-2}T\varphi^2(N)N^{-1}}$. If we select $N \asymp T$, we can deduce that:

$$R_T = \sum_{t=1}^{T} r_t = \mathcal{O}\left(\sqrt{T\gamma_T}\right). \tag{D.9}$$

$\square$

# E  PROOF OF THEOREM 4

In this section, we present the proof for Theorem 4.

*Proof of Theorem 4.* To derive an upper bound for $r_t$, we decompose it into two constituent terms as follows:

$$r_t = L\left(\lambda_t^+, \theta_{\lambda_t^+}^N\right) - L\left(\lambda^*, \theta_{\lambda^*}^*\right)$$
$$= \underbrace{L\left(\lambda_t^+, \theta_{\lambda_t^+}^N\right) - L\left(\lambda_t^+, \theta_{\lambda_t^+}^*\right)}_{\text{Term 11}} + \underbrace{L\left(\lambda_t^+, \theta_{\lambda_t^+}^*\right) - L\left(\lambda^*, \theta_{\lambda^*}^*\right)}_{\text{Term 12}}. \tag{E.1}$$

From inequality D.2, we can derive an upper bound for Term 11:

$$L\left(\lambda_t^+, \theta_{\lambda_t^+}^N\right) - L\left(\lambda_t^+, \theta_{\lambda_t^+}^*\right) \leq \alpha(N)N^{-\frac{1}{2}}. \tag{E.2}$$

Where $\alpha(N) = \mathcal{O}(\log^{3/2} N)$. Notice that at each round $t \geq 1$, based on the selection of $\lambda_t^+$ in Algorithm 2, i.e., $\lambda_t^+ = \text{argmax}_\lambda \left(-\mu_{t-1}(\lambda) + \beta_t\sigma_{t-1}(\lambda)\right)$. we have

$$-\mu_{t-1}(\lambda^*) + \beta_t\sigma_{t-1}(\lambda^*) \leq -\mu_{t-1}(\lambda_t^+) + \beta_t\sigma_{t-1}(\lambda_t^+).$$

From lemma 2, we have

$$-L\left(\lambda^*, \theta_{\lambda^*}^*\right) \leq -\mu_{t-1}(\lambda^*) + \beta_t\sigma_{t-1}(\lambda^*),$$
$$L\left(\lambda_t^+, \theta_{\lambda_t^+}^*\right) - \mu_{t-1}(\lambda_t^+) \leq \beta_t\sigma_{t-1}(\lambda_t^+).$$

Utilizing the above three inequalities, we can derive an upper bound for Term 12:

$$L\left(\lambda_t^+, \theta_{\lambda_t^+}^*\right) - L\left(\lambda^*, \theta_{\lambda^*}^*\right) \leq L\left(\lambda_t^+, \theta_{\lambda_t^+}^*\right) - \mu_{t-1}(\lambda^*) + \beta_t\sigma_{t-1}(\lambda^*)$$
$$\leq L\left(\lambda_t^+, \theta_{\lambda_t^+}^*\right) - \mu_{t-1}(\lambda_t^+) + \beta_t\sigma_{t-1}(\lambda_t^+)$$
$$\leq 2\beta_t\sigma_{t-1}(\lambda_t^+). \tag{E.3}$$

By combining the upper bound E.2 for Term 11 and the upper bound E.3 for Term 12 with Equation E.1, we obtain

$$R_T = \sum_{t=1}^{T} r_t \leq 2\beta_T \sum_{t=1}^{T} \sigma_{t-1}\left(\lambda_t^+\right) + T\alpha(N)N^{-\frac{1}{2}}.$$

From lemma 17, we have $\sum_{t=1}^{T} \sigma_{t-1}\left(\lambda_t^+\right) = \mathcal{O}\left(\sqrt{T\gamma_T}\right)$. Additionally, according to the definition, we have $\beta_T = \sqrt{B^2 + \sigma^{-2}T\varphi^2(N)N^{-1}}$. Hence with probability at least $1 - \delta$,

$$R_T = \mathcal{O}\left(\sqrt{(B^2 + TN^{-1})T\gamma_T} + TN^{-\frac{1}{2}}\right). \tag{E.4}$$

If we select $N \asymp T$, we can obtain that:

$$R_T = \mathcal{O}\left(\sqrt{T\gamma_T}\right). \tag{E.5}$$

$\square$

# F  EXTENSIONS

In this section, we present some extensions of our analyses. We consider two extensions: extension to non-convex scenarios and extension to non-smooth scenarios.

## F.1  EXTENSION TO NON-CONVEX SCENARIOS

In this section, we extend Theorem 1 to non-convex loss functions as follows.

**Theorem 18.** *Suppose that the function $\ell(\lambda, \theta, z)$ is $K$-Lipschitz and $\gamma$-smooth with respect to $\theta$, uniformly bounded by $M$, and also satisfies $\frac{1}{2}\|\nabla_\theta \ell(\lambda, \theta, z)\|^2 \geq \mu(\ell(\lambda, \theta, z) - \ell(\lambda, \theta_\lambda^*, z))$, $\mu > 0$. We perform SGD with step sizes $\eta_j = \frac{2j+1}{2\mu(j+1)^2}$ on a sample $S^{tr}$ drawn from the distribution $\mathcal{D}$ at the inner level. Let $S^{val} = \{z_i^{val}\}_{i=1}^m$ represent the validation set drawn from the distribution $\mathcal{D}$. Suppose $\gamma \leq N\mu/4$. Choose $N \asymp n \asymp m$. Then, with a probability of at least $1 - \delta$, we have:*

$$L^{val}(\lambda, \theta_\lambda^N, S^{val}) - L(\lambda, \theta_\lambda^*) = \mathcal{O}\left(\left(N^{-\frac{1}{2}}\mu^{-\frac{1}{2}} + N^{-\frac{1}{4}}\mu^{-\frac{3}{4}}\right)\log N\right)$$

*As a comparison, if $\ell(\lambda, \theta, z)$ is convex and $\eta_j \asymp N^{-\frac{1}{2}}$, then we have:*

$$L^{val}(\lambda, \theta_\lambda^N, S^{val}) - L(\lambda, \theta_\lambda^*) = \mathcal{O}\left(N^{-\frac{1}{2}}\log N\right)$$

*Proof of Theorem 18.* To establish an upper bound of the excess risk of the inner-level SGD, denoted as $L^{val}\left(\lambda, \theta_\lambda^N, S^{val}\right) - L\left(\lambda, \theta_\lambda^*\right)$, we introduce a highly effective decomposition method:

$$L^{val}\left(\lambda, \theta_\lambda^N, S^{val}\right) - L\left(\lambda, \theta_\lambda^*\right) = L^{val}\left(\lambda, \theta_\lambda^N, S^{val}\right) - L(\lambda, \theta_\lambda^N) + L(\lambda, \theta_\lambda^N) - L\left(\lambda, \theta_\lambda^*\right). \tag{F.1}$$

As elucidated in Remark 5 of Lei & Ying (2021), based on the pointwise hypothesis stability analysis and the optimization error bound in Karimi et al. (2016), it was shown with probability at least $1 - \delta$ (Charles & Papailiopoulos (2018)) that

$$L(\lambda, \theta_\lambda^N) - L\left(\lambda, \theta_\lambda^*\right) = \mathcal{O}\left(\frac{1}{\sqrt{n\mu\delta}} + \frac{1}{N^{\frac{1}{4}}\mu^{\frac{3}{4}}\delta^{\frac{1}{2}}}\right). \tag{F.2}$$

From the equation B.7 in the proof of Theorem 1 in Section B, we obtain

$$L\left(\lambda, \theta_\lambda^N\right) - L^{tr}\left(\lambda, \theta_\lambda^N, S^{tr}\right) = \mathcal{O}\left(\eta Nn^{-1}\log n \log\left(1/\delta\right) + M\sqrt{n^{-1}\log\left(1/\delta\right)}\right). \tag{F.3}$$

Selecting $N \asymp m \asymp n$, we obtain

$$L^{val}(\lambda, \theta_\lambda^N, S^{val}) - L(\lambda, \theta_\lambda^*) = \mathcal{O}\left(\left(N^{-\frac{1}{2}}\mu^{-\frac{1}{2}} + N^{-\frac{1}{4}}\mu^{-\frac{3}{4}}\right)\log N\right)$$

The proof is complete. $\square$

**Practical insights: Inner Unit Horizon as the Square of Outer-Level Iterations for Non-convex functions.** Unlike the case with convex, for the non-convex situation, the optimal number of inner-level SGD optimization iterations is $N \asymp T^2$. Hence, we attained the regret bound $R_T = \mathcal{O}(\sqrt{T\gamma_T})$. .

### F.2 EXTENSION TO NON-SMOOTH SCENARIOS

In this section, we extend Theorem 1 to non-smooth loss functions as follows. We remove the smoothness assumption by introducing an additional term $\eta\sqrt{N}$ in the excess risk bound. Moreover, we adopt alternative criteria, including weaker conditions, such as defining $\alpha$-Hölder continuity as:

**Definition 4.** Let $\gamma > 0$, $\alpha \in (0, 1]$. We say $\nabla_\theta \ell$ is $\alpha$-Hölder continuous if for any $\theta_1, \theta_2, z, \lambda$,

$$\|\nabla_\theta \ell(\lambda, \theta_1, z) - \nabla_\theta \ell(\lambda, \theta_2, z)\| \le \gamma \|\theta_1 - \theta_2\|_2^\alpha.$$

The theorem is updated as follows

**Theorem 19.** *Suppose that the function $\ell(\lambda, \theta, z)$ is $K$-Lipschitz continuous, and convex with respect to $\theta$, uniformly bounded by $M$. Let $p > \frac{1}{2}$. We perform SGD with step sizes $\eta_j = \eta \asymp N^{-p}$ on a sample $S^{tr}$ drawn from the distribution $\mathcal{D}$ at the inner level. Let $S^{val} = \{z_i^{val}\}_{i=1}^m$ represent the validation set drawn from the distribution $\mathcal{D}$. Choose $N \asymp n \asymp m$. Then, with a probability of at least $1 - \delta$, we have:*

$$L^{val}(\lambda, \theta_\lambda^N, S^{val}) - L(\lambda, \theta_\lambda^*) = \mathcal{O}\left(N^{\frac{1}{2}-p}\log(N) + N^{-p}\log(N) + N^{-\frac{1}{2}}\log(N)\right)$$

*Furthermore, if $\ell(\lambda, \theta, z)$ is $\alpha$-Hölder continuous, then we have:*

$$L^{val}(\lambda, \theta_\lambda^N, S^{val}) - L(\lambda, \theta_\lambda^*) = \mathcal{O}\left(N^{1-\frac{p}{1-\alpha}}\log(N) + N^{-p}\log(N) + N^{-\frac{1}{2}}\log(N)\right)$$

*As a comparison, if $\ell(\lambda, \theta, z)$ is smooth, then we have:*

$$L^{val}(\lambda, \theta_\lambda^N, S^{val}) - L(\lambda, \theta_\lambda^*) = \mathcal{O}\left(N^{-p}\log(N) + N^{-\frac{1}{2}}\log(N)\right)$$

*Proof of Theorem 19.* Similar to the proof of Theorem 1 in Section B, for establishing an upper bound on the excess risk $L^{val}(\lambda, \theta_\lambda^N, S^{val}) - L(\lambda, \theta_\lambda^*)$, we employ an effective decomposition method:

$$L^{val}\left(\lambda, \theta_\lambda^N, S^{val}\right) - L\left(\lambda, \theta_\lambda^*\right) = \underbrace{L^{val}\left(\lambda, \theta_\lambda^N, S^{val}\right) - L^{tr}\left(\lambda, \theta_\lambda^N, S^{tr}\right)}_{\text{Term 9}}$$

$$+ \underbrace{L^{tr}\left(\lambda, \theta_\lambda^N, S^{tr}\right) - L^{tr}\left(\lambda, \theta_\lambda^*, S^{tr}\right)}_{\text{Term 10}} + \underbrace{L^{tr}\left(\lambda, \theta_\lambda^*, S^{tr}\right) - L\left(\lambda, \theta_\lambda^*\right)}_{\text{Term 11}}.$$

Regarding Term 10 and Term 11, based on Lemma 10 and Lemma 11, we derive:

$$L^{tr}\left(\lambda, \theta_\lambda^N, S^{tr}\right) - L^{tr}\left(\lambda, \theta_\lambda^*, S^{tr}\right) + L^{tr}\left(\lambda, \theta_\lambda^*, S^{tr}\right) - L\left(\lambda, \theta_\lambda^*\right)$$

$$= \mathcal{O}\left((N^{-\frac{1}{2}} + N^{p-1})\log(N)\right)$$

$$= \mathcal{O}\left(N^{-\frac{1}{2}}\log(N)\right). \tag{F.4}$$

For Term 9, we decompose it as follows:

$$L^{val}\left(\lambda, \theta_\lambda^N, S^{val}\right) - L^{tr}\left(\lambda, \theta_\lambda^N, S^{tr}\right) = L^{val}\left(\lambda, \theta_\lambda^N, S^{val}\right) - L(\lambda, \theta_\lambda^N) + L(\lambda, \theta_\lambda^N) - L^{tr}\left(\lambda, \theta_\lambda^N, S^{tr}\right).$$

As demonstrated in B.8, we utilize Lemma 6 with $c = \frac{M}{m}$ to acquire, with a probability of at least $1 - \delta$,

$$|L^{val}\left(\lambda, \theta_\lambda^N, S^{val}\right) - L\left(\lambda, \theta_\lambda^N\right)| \le M\sqrt{\frac{1}{2m}\log(2/\delta)}. \tag{F.5}$$

Choose $N \asymp n \asymp m$, we obtain:

$$L^{val}\left(\lambda, \theta_\lambda^N, S^{val}\right) - L\left(\lambda, \theta_\lambda^N\right) = \mathcal{O}\left(N^{-\frac{1}{2}}\right). \tag{F.6}$$

Concerning the generalization error $L(\lambda, \theta_\lambda^N) - L^{tr}\left(\lambda, \theta_\lambda^N, S^{tr}\right)$, by combining Lemma 8 with Theorem 3.3 in Bassily et al. (2020), we obtain, with a probability of at least $1 - \delta$,

$$L(\lambda, \theta_\lambda^N) - L^{tr}\left(\lambda, \theta_\lambda^N, S^{tr}\right) = \mathcal{O}\left(N^{\frac{1}{2}-p}\log N + N^{-p}\log N + N^{-\frac{1}{2}}\right) \tag{F.7}$$

Then, we obtain:

$$L^{val}(\lambda, \theta_\lambda^N, S^{val}) - L(\lambda, \theta_\lambda^*) = \mathcal{O}\left(N^{\frac{1}{2}-p}\log(N) + N^{-p}\log(N) + N^{-\frac{1}{2}}\log(N)\right)$$

Furthermore, if $\ell(\lambda, \theta, z)$ is $\alpha$-Hölder continuous, then by combining Lemma 8 with Proposition G.1. in Lei & Ying (2020), we obtain, with a probability of at least $1 - \delta$,

$$L(\lambda, \theta_\lambda^N) - L^{tr}\left(\lambda, \theta_\lambda^N, S^{tr}\right) = \mathcal{O}\left(N^{1-\frac{p}{1-\alpha}}\log N + N^{-p}\log N + N^{-\frac{1}{2}}\right) \tag{F.8}$$

Then, we obtain:

$$L^{val}(\lambda, \theta_\lambda^N, S^{val}) - L(\lambda, \theta_\lambda^*) = \mathcal{O}\left(N^{1-\frac{p}{1-\alpha}}\log(N) + N^{-p}\log(N) + N^{-\frac{1}{2}}\log(N)\right).$$

The proof is complete. $\qquad\square$

