# OpenReview forum: "Convergence of Bayesian Bilevel Optimization"
_ICLR.cc/2024/Conference — ICLR 2024 spotlight_

### Official Review · Reviewer_1wgY · 2023-10-20

**Soundness:** 3 good
**Presentation:** 3 good
**Contribution:** 3 good
**Rating:** 8
**Confidence:** 2

**Summary:**

This paper provides a proof for the convergence of the Bayesian Bi-level optimization (BBO). By modeling the excess risk of the SGD-trained parameters, a regret bound is established for BBO with EI function, which bridges the analytical frameworks of Bayesian optimization and Bi-level optimization. Moreover, the authors introduce adaptable balancing coefficients to give a sublinear regret bound for BBO the UCB acquisition function.

**Strengths:**

1. The paper is theoretically solid. Useful regret bounds are provided and a convergence framework for BBO is established.

2. The paper is well-organized. The motivation, technique, proof schemes, and results are clearly stated.

3. Some tricks presented in the paper are interesting. For example, the adaptation of balancing coefficients could be a useful technique in other Bayesian applications.

**Weaknesses:**

1. The regularity assumptions are not intuitive. It would be better if the authors provided some real applications and models satisfying these assumptions.

2. Some assumptions are restrictive from the view of optimization, like the Lipschitz continuity and smoothness conditions in Theorem 1. Only a few classes of functions *simultaneously* satisfy them on $\mathrm{R}^d$.

**Questions:**

What is the fundamental difficulty of establishing convergence of BBO compared with other Bi-level algorithms?

**Details Of Ethics Concerns:**

Non.

---

> ### Author Response · Authors · 2023-11-22
> **Response 1**
>
> Thank you for your constructive comments and kind support! All your concerns have been carefully addressed as below. The manuscript is carefully revised accordingly. We sincerely hope our responses fully address your questions.
>
> **Q1: The regularity assumptions are not intuitive. It would be better if the authors provided some real applications and models satisfying these assumptions.**
>
> A: We appreciate this thought-provoking question. The RKHS norm $\\|f\\|_k=\sqrt{\langle f, f\rangle_k}$ of the objective function indeed provides a measure for its smoothness. We discuss several popular cases below:
>
> **(1) Gaussian process prior:**  the covariance function (or say kernel) dictates the smoothness of the sample paths. Specifically, a smaller RKHS norm indicates a smoother sample path. This relationship is showcased in the following example that the inequality $|f(x)-f(y)|=|\langle f, k(x, \cdot)-k(y, \cdot)\rangle| \leq\\|f\\|_k\\|k(x, \cdot)-k(y, \cdot)\\|_k$ stands for any function $f$ in the RKHS, as per the Cauchy-Schwarz inequality.
>
> **(2) Gaussian process with a Matérn kernel:** the parameter $\nu$ explicitly modulates smoothness. The Matérn covariance's general expression, $k_{\text {Matérn }}(d)=\frac{2^{1-\nu}}{\Gamma(\nu)}(\sqrt{2 \nu} d / \ell)^\nu K_\nu(\sqrt{2 \nu} d / \ell)$, illustrates that increasing $\nu$ enhances function smoothness.
>
> **(3) Sobolev space prior:**
> spaces $W^{k, p}(D)$ encompass functions with $k$-th order weak derivatives in $L^p(D)$. The order $k$ is a principal regulator of smoothness: higher $k$ values necessitate more integrable derivatives, thus ensuring more smoothness. For example, functions in $W^{2, p}(D)$ are "smoother" than those in $W^{1, p}(D)$ due to the requirements for second-order weak derivatives.
>
> **(4) Ridge regression:** the regularization term $\lambda\\|\beta\\|^2$ penalizes large parameter values, reflecting a bias towards parameters near zero. This acts as a "smoothness constraint", averting overfitting and promoting smooth and stable predictions. The regularization parameter $\lambda$ plays a crucial role in dictating the smoothness of the model's output. Specifically, a larger $\lambda$ leads to outcomes that are simpler and exhibit "more smoothness".
>
> Beyond Bayesian optimization and the agnostic GP bandit setting considered in this paper, the bounded RKHS norm stands across many other areas, including:
>
> **(1) Kernel Ridge Regression:** in the realm of kernel ridge regression, the boundedness of the RKHS norm is a prevalent hypothesis, pivotal for analyzing convergence properties ([1]).
>
> **(2) Recommender Systems:** a bounded RKHS norm plays a critical role, which helps manage model complexity effectively, thereby enhancing the practical performance of large-scale online recommendation systems ([2]).
>
> The bounded RKHS norm is a foundational assumption in numerous kernel methods and Gaussian process models, aiding in controlled learning and inference processes. Moreover, standard guess-and-doubling approaches are sufficient if no such bound is known in advance [3].

---

> ### Author Response · Authors · 2023-11-22
> **Response 2**
>
> **Q2: Some assumptions are restrictive from the view of optimization, like the Lipschitz continuity and smoothness conditions in Theorem 1. Only a few classes of functions simultaneously satisfy them on $R^d$.**
>
> A: Thanks. Practically, we would not use the whole $R^d$, and if exploding gradients do not happen, the two assumptions very likely hold. Also, following your suggestions, we relax the assumptions in our theory.
> We remove the smoothness assumption by introducing an additional term $\eta\sqrt{N}$ in the excess risk bound. In contrast, we use weaker conditions, such as $\alpha$-Hölder
> continuity defined as $\\|\nabla_{\theta} \ell(\lambda, \theta_1, z)-\nabla_{\theta} \ell(\lambda, \theta_2, z)\\|\leq \gamma\\|\theta_1-\theta_2\\|^{\alpha}_2$, $\alpha \in (0,1], \gamma > 0$ for any $\theta_1, \theta_2, z, \lambda$. The theorem is updated as follows:
>
>
> _**Theorem.** Suppose that the function $\ell(\lambda, \theta, z)$ is $K$-Lipschitz continuous, and convex with respect to $\theta$, uniformly bounded by $M$. We perform SGD with step sizes $\eta_j=\eta \asymp N^{-p}$ on a sample $S^{tr}$ drawn from the distribution $\mathcal{D}$ at the inner level. Let $S^{val}$ represent the validation set drawn from the distribution $\mathcal{D}$. Choose $N \asymp n \asymp m$. Then, with a probability of at least $1-\delta$, we have:_
> $$
> L^{val}(\lambda, \theta_{\lambda}^N, S^{val})-L(\lambda, \theta_{\lambda}^{\*})=\mathcal{O}\left(N^{\frac{1}{2}-p}\log(N)+N^{-p}\log(N)+N^{-\frac{1}{2}}\log(N)\right)
> $$
> _Furthermore, if $\ell(\lambda, \theta, z)$ is $\alpha$-Hölder continuous, then we have:_
> $$
> L^{val}(\lambda, \theta_{\lambda}^N, S^{val})-L(\lambda, \theta_{\lambda}^{\*})=\mathcal{O}\left(N^{1-\frac{p}{1-\alpha}}\log(N)+N^{-p}\log(N)+N^{-\frac{1}{2}}\log(N)\right)
> $$
> _As a comparison, if $\ell(\lambda, \theta, z)$ is smooth, then we have:_
> $$
> L^{val}(\lambda, \theta_{\lambda}^N, S^{val})-L(\lambda, \theta_{\lambda}^{\*})=\mathcal{O}\left(N^{-p}\log(N)+N^{-\frac{1}{2}}\log(N)\right)
> $$
>
> This discussion has been added to our paper; please kindly refer to  **Remark 1** on page four of the revised paper.
>
> **Q3: What is the fundamental difficulty of establishing convergence of BBO compared with other Bi-level algorithms?**
>
> A: Thank you for this insightful question. Establishing convergence guarantees for BBO poses unique challenges compared to other bilevel algorithms due to:
>
> **(1) Non-convexity and lack of derivatives**: In BBO, Bayesian optimization is utilized in the outer level to optimize hyperparameters, facing challenging non-convex, nonlinear, and potentially derivative-free scenarios. This deviates substantially from the more structured assumptions often made when analyzing other bilevel algorithms, significantly complicating the convergence analysis.
>
> **(2) Complex noise modeling**: BBO involves modeling the excess risk of inner-level SGD as noise in outer-level Bayesian optimization. This complex noise departs from simpler Gaussian noise assumptions commonly made in traditional bilevel problems, further increasing the difficulty of proving convergence.
>
> **(3) Integration of distinct methodologies**: BBO uniquely integrates SGD and Bayesian optimization, each with its own assumptions and theoretical challenges. This novel integration is not encountered in other bilevel algorithms and poses special difficulties in establishing unified convergence guarantees.
>
> In summary, BBO algorithms must address challenges such as non-convex, derivative-free hyperparameter tuning, SGD-induced noise, and the complex integration of Bayesian optimization with SGD, differentiating them from other bilevel methods. Traditional assumptions like convexity, simple noise models, and unified methodologies do not apply. Overcoming these distinctive difficulties is a key contribution of our theoretical analysis of BBO convergence.
>
> *Reference:*
>
> $\left[1\right]$ Zhang, H., Li, Y., Lu, W. On the Optimality of Misspecified Kernel Ridge Regression. Proceedings of the 40th International Conference on Machine Learning, 2023.
>
> $\left[2\right]$ Vanchinathan H P, Nikolic I, De Bona F, et al. Explore-exploit in top-n recommender systems via gaussian processes[C] Proceedings of the 8th ACM Conference on Recommender systems. 2014.
>
> $\left[3\right]$ Srinivas N, Krause A, Kakade S M, et al. Gaussian process optimization in the bandit setting: No regret and experimental design[J]. arXiv preprint 2009.
>
> $\left[4\right]$ Li S, Liu Y. High probability generalization bounds with fast rates for minimax problems[C] International Conference on Learning Representations. 2021.
>
> $\left[5\right]$ Fu S, Lei Y, Cao Q, et al. Sharper Bounds for Uniformly Stable Algorithms with Stationary Mixing Process[C] The Eleventh International Conference on Learning Representations. 2022.
>
> $\left[6\right]$ Klochkov Y, Zhivotovskiy N. Stability and Deviation Optimal Risk Bounds with Convergence Rate $ O (1/n) $[J]. Advances in Neural Information Processing Systems, 2021.

---

### Official Review · Reviewer_PGhy · 2023-10-30

**Soundness:** 2 fair
**Presentation:** 2 fair
**Contribution:** 2 fair
**Rating:** 6
**Confidence:** 1

**Summary:**

This paper introduces the initial theoretical assurance for Bayesian bilevel optimization (BBO). It is proved sublinear regret bounds suggest simultaneous convergence of the inner-level model parameters and outer-level hyperparameters to optimal configurations for generalization capability.

**Strengths:**

1. This work conducts lots of theoretical analysis Bayesian bilevel optimization (BBO). Specifically, a novel theoretical analysis of convergence guarantees for generalization performance within a BBO framework is provided.

2. A regret bound for BBO using the EI function is discussed in this work.

3. A significant advancement in this research lies in the conceptualization of SGD excess risk as a form of noise within the framework of Bayesian optimization. This approach allows for the adjustment of noise assumptions to better match real-world scenarios and greatly simplifies convergence analysis.

**Weaknesses:**

I can't find any experimental results in this work. I understand this work puts more attention on the theoretical analysis in Bayesian bilevel optimization (BBO). However, the authors should also conduct experiments to substantiate the theoretical analysis.

**Questions:**

My concerns are about the experimental results.

---

> ### Author Response · Authors · 2023-11-22
> **Response**
>
> Thank you for your constructive comments and kind support! All your concerns have been carefully addressed as below. We sincerely hope our responses fully address your questions.
>
> **Q: I can't find any experimental results in this work.**
>
> A: Thanks. Following your suggestions, we conducted numerical results during the rebuttal session, as reported below. Supplementary to our results, several published experiments in the literature [1], [2], [3] empirically demonstrate the convergence properties of BBO, validating our theory. We promise to add more experiments afterwards. We would also like to note that we are focused on understanding the theoretical principle of BBO.
>
> In the inner level, we employ momentum-based SGD to train a CNN with two convolutional layers and one fully connected layer on the MNIST dataset. In the outer level, Bayesian Optimization uses the EI acquisition function to adjust hyperparameters like the learning rate and the momentum coefficient. We fix the number of iterations for the outer-level BO and compare the number of iterations for the inner-level SGD under different scenarios, along with their respective convergence outcomes, as detailed in the table below.
>
> (1) Setting the number of outer Bayesian optimization steps as 20.
>
> | SGD iterations steps| 100 | 500 | 1000 | 2000 | 3000 |
> |----|----|----|----|----|----|
> |**Convergence properties (loss)**| $3.15 \pm 0.75$|$2.73 \pm 0.03$|$2.70 \pm 0.04$|$2.43 \pm 0.12$|$2.46 \pm 0.13$|
>
> (2) Setting the number of outer Bayesian optimization steps as 40.
>
> | SGD iterations steps| 500 | 1000 | 2000 | 3000 | 4000 |
> |----|----|----|----|----|----|
> |**Convergence properties (loss)**| $3.45\pm 0.85$|$2.42\pm 0.23$|$2.44\pm 0.55$|$2.39\pm 0.09$|$2.51\pm 0.09$|
>
> (3) Setting the number of outer Bayesian optimization steps as 60.
>
> | SGD iterations steps| 1000 | 2000 | 3000 | 4000 | 6000 |
> |----|----|----|----|----|----|
> |**Convergence properties (loss)**| $2.58 \pm 0.58$|$2.40 \pm 0.35$|$2.34 \pm 0.28$|$2.10 \pm 0.08$|$2.25 \pm 0.17$|
>
> The experiments are aligned with our theoretical analysis. Fixing the Bayesian optimization's iteration number, the loss function decreases as the SGD steps rise initially, while suboptimal hyperparameters cause high loss. Specifically, with only 20 outer-level iterations, excessively low tuning led to high loss even at high SGD steps. For 40 outer-level iterations and over 1000 SGD steps, we see signs of overtraining and inadequate tuning. For 60-step outer-level iterations, the loss at over 4000 SGD steps is less than in the 40-step setting, due to more adequate tuning. Yet, insufficient tuning still occurs at 6000 SGD steps under the 60-step setting.

---

> > ### Comment · Reviewer_PGhy · 2023-11-22
> >
> > Thanks for your responses. My concerns have been addressed and I'm happy to maintain the score as 6.

---

> > > ### Author Response · Authors · 2023-11-22
> > > **Thank you**
> > >
> > > Thank you for your support!

---

### Official Review · Reviewer_66Ye · 2023-10-30

**Soundness:** 3 good
**Presentation:** 3 good
**Contribution:** 2 fair
**Rating:** 6
**Confidence:** 3

**Summary:**

The paper focuses on Bayesian bilevel optimization (BBO), which combines outer-level Bayesian opti- mization for hyperparameter tuning with inner-level stochastic gradient descent (SGD) for model parameter optimization. The paper proves sublinear regret bounds for BBO using expected improvement (EI) and upper confidence bound (UCB) acquisitions. This provides theoretical assurance that BBO enables simul- taneous convergence of parameters and hyperparameters. For EI, the optimal number of SGD iterations is shown to be N ≍ T 2, balancing training and tuning. This achieves regret savings compared to previous works. For UCB, sublinear regret is proven even with fewer SGD iterations, showing UCB is more robust to SGD noise. The UCB balancing coeﬀicients are adapted based on the SGD/Bayesian iteration ratio. The analysis bridges the gap between Bayesian and bilevel optimization frameworks by modeling SGD excess risk, which enables adapting convergence guarantees to the BBO setting.

**Strengths:**

(1) The paper provides a new theoretical analysis bridging the frameworks of Bayesian optimization and bilevel optimization by modeling the excess risk of SGD-trained parameters as noise to tackle challenges in convergence guarantees for BBO generalization performance.

(2) Based on a noise assumption better suited to practical situations, the authors derive sublinear regret bounds for Bayesian bilevel optimization using the expected improvement function, which is better than previous work.

(3) By introducing adaptable balancing coeﬀicients $\beta_t$ for the UCB acquisition function, the paper establishes a sublinear regret bound for BBO with UCB that holds with fewer SGD steps, enhancing inner unit horizon flexibility and overcoming limitations of rapidly increasing coeﬀicients from previous analyses.

**Weaknesses:**

(1) The current paper is primarily theoretical with a lack of numerical experiments on actual data, which limits the persuasiveness. Experiments using real-world hyperparameter tuning tasks could offer tangible evidence of the convergence behavior and help assess how well the assumptions fit such scenarios.

(2) This paper focuses solely on Gaussian process priors for the Bayesian optimization portion, but the choice of prior may significantly impact the convergence guarantees. The current analysis leverages nice properties of GP priors and posters but may not directly extend to other priors that require different proof techniques, which could limit wider applicability.

(3) Bayesian optimization is adopted for hyperparameter tuning at the outer layer, so the algorithm in this paper may require extensive sampling and integration to estimate the posterior distribution, making it computationally demanding and diﬀicult to apply to high-dimensional complex problems.

**Questions:**

(1) How do the convergence guarantees extend to deep neural network training? Are there any unique challenges posed by DNNs?

(2) For the inner-level SGD, will using SVRG or introducing acceleration techniques lead to better corresponding results compared to standard SGD?

(3) Do assumptions such as the bounded RKHS norm of the objective function correspond cleanly to properties of other priors?

---

> ### Author Response · Authors · 2023-11-22
> **Response 1**
>
> Thank you for your constructive comments and kind support! All your concerns have been carefully addressed as below. The manuscript is carefully revised accordingly. We sincerely hope our responses fully address your questions.
>
> **Q1: How do the convergence guarantees extend to deep neural network training? Are there any unique challenges posed by DNNs?**
>
> A: Thank you for this valuable feedback. The primary challenge in extending our convergence guarantees to DNNs stems from their highly non-convex nature. To address this, we have expanded our theoretical analysis to encompass non-convex functions as well. Specifically, we have established the following theorem :
>
>
> **Theorem.** Suppose that the function $\ell(\lambda, \theta, z)$ is $K$-Lipschitz  and $\gamma$-smooth with respect to $\theta$, uniformly bounded by $M$,  and also satisfies $\frac{1}{2} \\| \nabla_\theta\ell(\lambda,\theta,z)\\|^2\geq \mu(\ell(\lambda,\theta,z)-\ell(\lambda,\theta_{\lambda}^*,z))$, $\mu>0$. We perform SGD with step sizes $\eta_j=\frac{2j+1}{2\mu(j+1)^2}$ on a sample $S^{tr}$ drawn from the distribution $\mathcal{D}$ at the inner level. Let $S^{val}$ represent the validation set drawn from the distribution $\mathcal{D}$. Suppose $\gamma\leq N\mu /4 $. Choose $N \asymp n \asymp m$. Then, with a probability of at least $1-\delta$, we have:
> $$
> L^{val}(\lambda, \theta_{\lambda}^N, S^{val})-L(\lambda, \theta_{\lambda}^*)=\mathcal{O}\left(\left(N^{-\frac{1}{2}}\mu^{-\frac{1}{2}}+N^{-\frac{1}{4}}\mu^{-\frac{3}{4}}\right)\log N\right).
> $$
> As a comparison, if $\ell(\lambda, \theta, z)$ is convex and $\eta_j \asymp N^{-\frac{1}{2}}$, then we have:
> $$
> L^{val}(\lambda, \theta_{\lambda}^N, S^{val})-L(\lambda, \theta_{\lambda}^*)=\mathcal{O}\left(N^{-\frac{1}{2}}\log N\right).
> $$
>
> We believe that it is very necessary to conduct more extensive research in the direction of DNNs, and we will continue to explore this in our future work.
>
>
> **Q2: For the inner-level SGD, will using SVRG or introducing acceleration techniques lead to better corresponding results compared to standard SGD?**
>
> A: Thanks for the insightful question.
> Employing SVRG in the inner level will yield a faster convergence rate and a smaller excess risk.
>
> **Convergence rate:** Our theory suggests that the inner-level excess risk is the noise in the outer-level optimization. Faster inner-level optimization naturally implies faster convergence of the whole BBO. Specifically, the regret of BBO can be decomposed as follows:
> $$
> r_{t}=L\left(\lambda_{t},\theta_{\lambda_{t}}^{N}\right)-L\left(\lambda^{\*},\theta_{\lambda^{\*}}^{\*}\right)=L\left(\lambda_{t},\theta_{\lambda_{t}}^{N}\right)-L^{val}(\lambda_{t},\theta_{\lambda_{t}}^{N},S^{val})+L^{val}(\lambda_{t},\theta_{\lambda_{t}}^{N},S^{val})-L\left(\lambda_{t},\theta_{\lambda_{t}}^{\*}\right)+L\left(\lambda_{t},\theta_{\lambda_{t}}^{\*}\right)-L\left(\lambda^{\*},\theta_{\lambda^{\*}}^{\*}\right).
> $$
> SGD/SVRG influences the inner-level excess risk term $L^{val}(\lambda_{t},\theta_{\lambda_{t}}^{N},S^{val})-L\left(\lambda_{t},\theta_{\lambda_{t}}^{\*}\right)$, and further the estimation risk term in the outer level: $L\left(\lambda_{t},\theta_{\lambda_{t}}^{\*}\right)-L\left(\lambda^{\*},\theta_{\lambda^{\*}}^{\*}\right)$.
>
> **Excess risk:** The inner-level excess risk term can be decomposed as follows:
> $$
> \begin{aligned}
>  L^{v a l}\left(\lambda_t, \theta_{\lambda_t}^N, S^{v a l}\right)-L\left(\lambda_t, \theta_{\lambda_t}^{\*}\right)
> = L^{v a l}\left(\lambda_t, \theta_{\lambda_t}^N, S^{v a l}\right)-L^{t r}\left(\lambda_t, \theta_{\lambda_t}^N, S^{t r}\right)+L^{t r}\left(\lambda_t, \theta_{\lambda_t}^N, S^{t r}\right)-L^{t r}\left(\lambda_t, \theta_{\lambda_t}^{\*}, S^{t r}\right)+L^{t r}\left(\lambda_t, \theta_{\lambda_t}^{\*}, S^{t r}\right)-L\left(\lambda_t, \theta_{\lambda_t}^{\*}\right) .
> \end{aligned}
> $$
> SGD/SVRG influences both the optimization error term $L^{t r}\left(\lambda_t, \theta_{\lambda_t}^N, S^{t r}\right)-L^{t r}\left(\lambda_t, \theta_{\lambda_t}^{\*}, S^{t r}\right)$ and the generalization gap term $L^{v a l}\left(\lambda_t, \theta_{\lambda_t}^N, S^{v a l}\right)-L^{t r}\left(\lambda_t, \theta_{\lambda_t}^N, S^{t r}\right)$. [1] suggests that SVRG exhibits a lower optimization error compared to SGD. [2] and [3] suggest that SVRG also exhibits better generalizability.
> Combined with these results, our theory shows that the excess risk of SVRG is lower than SGD, consequently reducing the excess risk of BBO.

---

> ### Author Response · Authors · 2023-11-22
> **Response 2**
>
> **Q3: Do assumptions such as the bounded RKHS norm of the objective function correspond cleanly to properties of other priors?**
>
> A: We appreciate this thought-provoking question. The RKHS norm $\\|f\\|_k=\sqrt{\langle f, f\rangle_k}$ of the objective function indeed provides a measure for its smoothness. We discuss several popular cases below:
>
> **(1) Gaussian process prior:**  the covariance function (or say kernel) dictates the smoothness of the sample paths. Specifically, a smaller RKHS norm indicates a smoother sample path. This relationship is showcased in the following example that the inequality $|f(x)-f(y)|=|\langle f, k(x, \cdot)-k(y, \cdot)\rangle| \leq\\|f\\|_k\\|k(x, \cdot)-k(y, \cdot)\\|_k$ stands for any function $f$ in the RKHS, as per the Cauchy-Schwarz inequality.
>
> **(2) Gaussian process with a Matérn kernel:** the parameter $\nu$ explicitly modulates smoothness. The Matérn covariance's general expression, $k_{\text {Matérn }}(d)=\frac{2^{1-\nu}}{\Gamma(\nu)}(\sqrt{2 \nu} d / \ell)^\nu K_\nu(\sqrt{2 \nu} d / \ell)$, illustrates that increasing $\nu$ enhances function smoothness.
>
> **(3) Sobolev space prior:**
> spaces $W^{k, p}(D)$ encompass functions with $k$-th order weak derivatives in $L^p(D)$. The order $k$ is a principal regulator of smoothness: higher $k$ values necessitate more integrable derivatives, thus ensuring more smoothness. For example, functions in $W^{2, p}(D)$ are "smoother" than those in $W^{1, p}(D)$ due to the requirements for second-order weak derivatives.
>
> **(4) Ridge regression:** the regularization term $\lambda\\|\beta\\|^2$ penalizes large parameter values, reflecting a bias towards parameters near zero. This acts as a "smoothness constraint", averting overfitting and promoting smooth and stable predictions. The regularization parameter $\lambda$ plays a crucial role in dictating the smoothness of the model's output. Specifically, a larger $\lambda$ leads to outcomes that are simpler and exhibit "more smoothness".
>
> These discussions have been added to our paper; please kindly refer to **Section 3.2.1** on page four.
>
> *Reference:*
>
> $\left[1\right]$ Reddi S J, Hefny A, Sra S, et al. Stochastic variance reduction for nonconvex optimization[C] International conference on machine learning. PMLR, 2016: 314-323.
>
> $\left[2\right]$ Meng Q, Wang Y, Chen W, et al. Generalization error bounds for optimization algorithms via stability[C] Proceedings of the AAAI Conference on Artificial Intelligence. 2017, 31(1).
>
>
> $\left[3\right]$ Lei Y, Ying Y. Sharper generalization bounds for learning with gradient-dominated objective functions[C]//International Conference on Learning Representations. 2020.

---

### Official Review · Reviewer_pR2Q · 2023-10-31

**Soundness:** 4 excellent
**Presentation:** 4 excellent
**Contribution:** 3 good
**Rating:** 6
**Confidence:** 5

**Summary:**

This is paper presents the first convergence analysis of Bayesian bilevel optimization where the outer level is hyperparameter tuning and the inner level is SGD. The key results are sublinear regret bounds showing the convergence behaviors of both outer and inner optimization problems. The key technical novelty is modeling the excess risk of SGD training as the noise of the outer Bayesian optimization. This paper doesn’t have experiments.

**Strengths:**

1. First convergence analysis of BBO is important, which is the main contribution of this paper.
2. I appreciate the innovation that modeling the excess risk of inner level SGD-trained parameters as the primary noise source of outer-level BO. It makes great sense in this problem setting.
3. I like “practical insights” sections, which are helpful.
4. The whole paper is well written and easy to follow except some notation problems mentioned below.

**Weaknesses:**

1. Motivation of BBO is not very clear. No detail is shown in “significant promise in engineering applications” in 2nd paragraph of Introduction.
2. Convexity assumption in Definition 1 is strong. How can you assume the loss function is convex given potentially non-convex objective function? I want to learn more justification from the author.
3. L is taken as both loss function and Lipschitz constant, which introduces some confusion.
4. Upper bound in Theorem 1 is too vague, only showing dependence on N. How does it depend on other terms?

**Questions:**

1. Why is modeling the noise as a martingale difference a key limitation? Why does this approach not align with hyperparameter optimization? See fourth line of page 2.
2. In third line of Section 3.2, you assume function L has a uniquely determined value for each \lambda. In my understanding, it is needed otherwise some \theta rather than \theta* may lead to lower value given some \lambda and it would be hard to define regret. However, do you have more justification on this assumption especially in practical scenarios?
3. What’s $\varphi(N)$ in Theorem 1?

---

> ### Author Response · Authors · 2023-11-22
> **Response 1**
>
> Thank you for your constructive comments and kind support! All your concerns have been carefully addressed as below. The manuscript is carefully revised accordingly. We sincerely hope our responses fully address your questions.
>
> **Q1: Motivation of BBO is not very clear. No detail is shown in “significant promise in engineering applications” in 2nd paragraph of Introduction.**
>
> A: Thanks and addressed. As shown in [1], [2], [3], and [4], BBO has significant promise in training neural networks -- SGD trains a neural network in the inner level, while Bayesian optimization tunes critical hyperparameters (e.g., learning rates, hidden layer widths, and dropout probabilities) in the outer level. A detailed discussion has been added to our paper. Please kindly refer to the second paragraph of the **Introduction** section.
>
> **Q2: Convexity assumption in Definition 1 is strong. How can you assume the loss function is convex given potentially non-convex objective function? I want to learn more justification from the author.**
>
> A: Thanks. Following your suggestion, we have extended **Theorem 1** to non-convex loss functions as follows.
>
>
> **Theorem.** Suppose that the function $\ell(\lambda, \theta, z)$ is $K$-Lipschitz  and $\gamma$-smooth with respect to $\theta$, uniformly bounded by $M$,  and also satisfies $\frac{1}{2} \\| \nabla_\theta\ell(\lambda,\theta,z)\\|^2\geq \mu(\ell(\lambda,\theta,z)-\ell(\lambda,\theta_{\lambda}^{\*},z))$, $\mu>0$.  We perform SGD with step sizes $\eta_j=\frac{2j+1}{2\mu(j+1)^2}$ on a sample $S^{tr}$ drawn from the distribution $\mathcal{D}$ at the inner level. Let $S^{val}$ represent the validation set drawn from the distribution $\mathcal{D}$. Suppose $\gamma\leq N\mu /4 $. Choose $N \asymp n \asymp m$. Then, with a probability of at least $1-\delta$, we have:
> $$
> L^{val}(\lambda, \theta_{\lambda}^N, S^{val})-L(\lambda, \theta_{\lambda}^{\*})=\mathcal{O}\left(\left(N^{-\frac{1}{2}}\mu^{-\frac{1}{2}}+N^{-\frac{1}{4}}\mu^{-\frac{3}{4}}\right)\log N\right).
> $$
> As a comparison, if $\ell(\lambda, \theta, z)$ is convex and $\eta_j \asymp N^{-\frac{1}{2}}$, then we have:
> $$
> L^{val}(\lambda, \theta_{\lambda}^N, S^{val})-L(\lambda, \theta_{\lambda}^{\*})=\mathcal{O}\left(N^{-\frac{1}{2}}\log N\right).
> $$
> Our paper has been updated accordingly. Please kindly refer to **Reamrk 1** on page four. We believe that further research in this non-convex aspect is necessary, and we will continue to explore it in future work.
>
> **Q3: What’s $\varphi(N)$ in Theorem 1? Upper bound in Theorem 1 is too vague, only showing dependence on N. How does it depend on other terms?**
>
> A: Thanks. $\varphi(N)$ is defined as follows.
> $$\varphi(N) = \mathcal{O}\left(K^2 \log N \log (1 / \delta)+M \sqrt{\log (1 / \delta)}+M \sqrt{\log (2 / \delta)}+ \log ^{3/2}(N/\delta)\right).$$
> Despite the SGD iteration number $N$ being the most crucial and discussed factor, the excess risk of SGD also depends on the Lipschitz constant $K$ of loss $\ell(\lambda,\theta,z)$ with respect to $\theta$, the upper bound $M$ of loss function, step size $\eta$ of SGD, training sample size $n$, and validation sample size $m$. Specifically, with the probability at least $1-\delta$, the excess risk bound is as follows,
> $$
> \begin{aligned}
> L^{v a l}\left(\lambda, \theta_\lambda^N, S^{v a l}\right)-L\left(\lambda, \theta_\lambda^{\*}\right)=\mathcal{O}\left(K^2 \eta N n^{-1} \log n \log (1 / \delta)+M \sqrt{n^{-1} \log (1 / \delta)}+M \sqrt{m^{-1} \log (2 / \delta)}+N^{-1/2} \log ^{3/2}(N/\delta)\right).
> \end{aligned}
> $$
> This is the ``full'' order showing the dependence on all factors.
>
>
> We have incorporated this discussion in the revised **Theorem 1**. The proof is also updated accordingly.

---

> ### Author Response · Authors · 2023-11-22
> **Response 2**
>
> **Q4: Why is modeling the noise as a martingale difference a key limitation? Why does this approach not align with hyperparameter optimization? See fourth line of page 2.**
>
> A: Thanks. We respectively show below the noise sequence is not a martingale difference sequence.
>
> Firstly, recall that hyperparameter optimization aims to find $\lambda^{\*}=\underset{\lambda \in \Lambda}{\arg \min }\  L\left(\lambda, \theta_\lambda^{\*}\right)$, where $L(\lambda, \theta_\lambda^{\*})=E_{z \sim \mathcal{D}}[\ell(\lambda, \theta_\lambda^{\*}, z)]$ is the generalization error here. The noise is $\varepsilon_t=L^{v a l}\left(\lambda_t, \theta_{\lambda_t}^N, S^{v a l}\right)-L\left(\lambda_t, \theta_{\lambda_t}^{\*}\right)$, where  $\theta_{\lambda_t}^{\*}=\underset{\theta \in \Theta}{\arg \min }\ L(\lambda_t, \theta)$ and $\theta_{\lambda_t}^N$ is obtained via SGD. The conditional expectation $E\left[\varepsilon_t \mid \varepsilon_{<t}\right]=0$ does not hold here. This is because $\theta_{\lambda_t}^N$ depends on the training set $S^{tr}$ while $\theta_{\lambda_t}^{\*}$ does not. Therefore, the noise sequence is not a martingale difference sequence.
>
> If the generalization gap is defined as $L^{tr}\left(\lambda_t, \theta_{\lambda_t}^N, S^{tr}\right)-L\left(\lambda_t, \theta_{\lambda_t}^N\right)$, the martingale difference sequence assumption still does not hold since $\theta_{\lambda_t}^N$ relies on $S^{tr}$.
>
> This discussion has been added to our paper; please kindly refer to the **Section 4.2.1** section on page 8.
>
> **Q5: In third line of Section 3.2, you assume function $L$ has a uniquely determined value for each $\lambda$. In my understanding, it is needed otherwise some $\theta$ rather than $\theta^{\*}$ may lead to lower value given some $\lambda$ and it would be hard to define regret. However, do you have more justification on this assumption especially in practical scenarios?**
>
> A: Thanks. We appreciate the chance to clarify. Hope the concern can be cleared.
>
> When $\lambda$ is given, $\theta_\lambda^{\*}=\underset{\theta \in \Theta}{\arg \min }\ L(\lambda, \theta)$ could be a set, but for any $\theta_\lambda^{\*}$ in $\underset{\theta \in \Theta}{\arg \min }\ L(\lambda, \theta)$, the loss function $L(\lambda, \theta_\lambda^{\*})$ has a unique value $\underset{\theta \in \Theta}{\min }\ L(\lambda, \theta)$. This stands generally.
>
> This discussion has been carefully added to our paper, please kindly refer to the **Section 3.2** on page four.
>
> *Reference:*
>
> $\left[1\right]$ Nguyen V, Schulze S, Osborne M. Bayesian optimization for iterative learning[J]. Advances in Neural Information Processing Systems, 2020, 33: 9361-9371.
>
> $\left[2\right]$ Snoek J, Rippel O, Swersky K, et al. Scalable bayesian optimization using deep neural networks[C]//International conference on machine learning. PMLR, 2015: 2171-2180.
>
> $\left[3\right]$ Dewancker I, McCourt M, Clark S. Bayesian optimization for machine learning: A practical guidebook[J]. arXiv preprint arXiv:1612.04858, 2016.
>
> $\left[4\right]$ Gelbart M A, Snoek J, Adams R P. Bayesian optimization with unknown constraints[C]//Proceedings of the Thirtieth Conference on Uncertainty in Artificial Intelligence. 2014: 250-259.
>
> $\left[5\right]$ Liu H, Simonyan K, Yang Y. DARTS: Differentiable Architecture Search[C]//International Conference on Learning Representations. 2018.

---

> > ### Comment · Reviewer_pR2Q · 2023-11-23
> >
> > Thank you so much for your clarification. My rating remains a positive one as before.

---

### Meta-Review · Area_Chair_RtQN · 2023-12-06

**Metareview:**

This paper presents a convergence analysis of Bayesian bilevel optimization. It is claimed that this is the first theoretical guarantee result, proving sublinear regret bounds. All the reviewers agree that the paper provides solid results, which are deserved to be presented in ICLR. One critical concern is that the paper does not contain any experimental results that support the theoretical claims. In the rebuttal period, the authors provided several published experimental results, showing that they are aligned with the theoretical results in the paper. The authors promised to add more experiments. After the rebuttal, all reviewers maintained their initial decisions, supporting this work.

**Justification For Why Not Higher Score:**

Experiments should be added to justify the theoretical results.

**Justification For Why Not Lower Score:**

This paper presents the first convergence analysis of Bayesian bilevel optimization.

---

### Decision · Program_Chairs · 2024-01-16

Accept (spotlight)